# Measurements and Calculations of Enhanced Side/Back Scattering of Visible Radiation by Black Carbon Aggregates

Carynelisa Haspel[1], Cuiqi Zhang[2], Martin J. Wolf[3], Daniel J. Cziczo[4,5], Maor Sela[6]

[1]The Fredy and Nadine Herrmann Institute of Earth Sciences, The Hebrew University of Jerusalem, Jerusalem 9190401, Israel

[2]Institute for Atmospheric and Climate Science, ETH Zürich, Zürich, 8092, Switzerland

[3]Yale School of the Environment, Yale University, New Haven, CT 06511, USA

[4]Department of Earth, Atmospheric, and Planetary Sciences, Purdue University, West Lafayette, IN 47907, USA

[5]Department of Earth, Atmospheric, and Planetary Sciences, Massachusetts Institute of Technology, Cambridge, MA 02139, USA

[6]Atmospheric, Oceanic and Planetary Physics, Department of Physics, University of Oxford, Oxford, UK

*Correspondence to*: Carynelisa Haspel (carynelisa.haspel@mail.huji.ac.il)

**Abstract.** Aerosol particles have both natural and anthropogenic origins and are ubiquitous in the atmosphere. One particularly important type is carbonaceous aerosol, including a specific subset, often termed 'elemental carbon' chemically or 'black carbon' (BC) radiatively. Carbonaceous aerosol particles have implications for atmospheric chemistry, human health, and climate both directly and via their ability to act as sites of cloud droplet or ice crystal formation. Laboratory experiments and theory are needed to better understand these particles, specifically their radiative impact. We present here laboratory measurements of scattering of visible radiation by analogues of atmospheric BC aggregates at scattering angles of 135±20° obtained using a depolarizing optical particle counter and accompanying theoretical calculations of scattering by compact and fractal theoretical BC aggregates. We show that with random orientation, the theoretical calculations reproduce the qualitative behavior of the measurements but are unable to reproduce the highest values of the linear depolarization ratio; we are only able to obtain high values of the linear depolarization ratio using fixed orientation. Both our measurements and our theoretical calculations point to the possibility that fresh/unaged/bare/uncoated BC aggregates, as opposed to the aged/coated BC or soot that was investigated in previous studies, can exhibit higher backscattering linear depolarization than previously assumed.

## 1 Introduction

Accurate calculations of the single scattering properties of black carbon (BC) aerosol particles are important for estimating their radiative forcing of climate and for interpreting remote sensing observations, and indeed many previous studies have been dedicated to this topic. See, for example, the excellent review by Kahnert and Kanngießer (2020). These single scattering properties include the scattering, absorption, and extinction cross section, and the scattering asymmetry factor or full scattering phase matrix. BC particles are often found in the atmosphere in the form of aggregates of primary particles,

and the aggregates are often described using fractal parameters. See, for example, Sorensen (2001). More extended fractal aggregates are generally considered to be analogues of relatively fresh/unaged black carbon (BC), while more compact, roughly spherical aggregates are considered to be analogues for BC that has "collapsed" into a quasi-spherical structure after cloud processing or aging (Ma et al., 2013; Sedlaceck et al., 2015). Thus, a proper calculation of the radiative properties of BC particles must include a proper description of the aggregate structure (see, e.g., Bond and Bergstrom (2006), Kahnert and Kanngießer (2020), and references therein). The aggregate shape of BC particles also causes linearly polarized incident light to become partially depolarized upon scattering (Lu and Sorensen, 1994; Bescond et al., 2013; Paulien et al., 2019). Thus, the linear depolarization ratio (the ratio of cross-polarized scattered intensity to incident intensity) can be a useful quantity for assessing aggregate shape effects in forward calculations and conversely for detecting the presence of aggregate-shaped particles and other non-spherical particles using remote sensing data.

Sela and Haspel (2021) presented theoretical calculations of scattering of visible radiation by pairs of aggregates comprised of spherical nano-scale primary particles. Each aggregate pair consisted of an ordered aggregate with a simple cubic (SC) configuration and a disordered aggregate with an ideal amorphous solid (IAS) configuration based on the model of Stachurski (2003; 2011; 2013), and the scattering was computed using the multiple sphere $T$-matrix (MSTM) model of Mackowski and Mishchenko (1996). Sela and Haspel (2021) found that holding all other parameters constant, in most cases, the overall scattering and absorption and hence extinction of radiation by ordered aggregates is stronger than for disordered aggregates. At the same time, they found that holding all other parameters constant, disordered aggregates tend to side scatter and back scatter more strongly than ordered aggregates.

To further investigate the influence of the configuration of the primary particles in an aggregate on side/back scattering by the aggregate, in the present study, we compare new theoretical calculations of the scattering of visible radiation by aggregates against scattering measurements conducted on analogues of atmospheric BC aggregates whose microphysical and ice nucleation properties were presented in Zhang et al. (2020). The BC sample sets labeled "COJ300" and "R2500U 400 nm" in Zhang et al. (2020) exhibit similar primary particle diameters ( $d_{\mathrm{pp}}$ ; ~35±10 nm) and mobility diameters ( $D_{\mathrm{m}}$ ; 400 nm) to one another. At the same time, the outer envelopes of the COJ300 samples appear more spherical, while the outer envelopes of the R2500U samples appear more extended (see Fig. 1). This is consistent with the fact that the mean fractal dimension ( $D_{\mathrm{f}}$ ) of the COJ300 samples (2.34 with a 95% confidence interval range of 2.12-2.56) is higher than the mean fractal dimension of the R2500U samples (1.92 with a 95% confidence interval range of 1.68-2.16). See Table 1 of Zhang et al. (2020). See also DeCarlo et al. (2004) for a comprehensive discussion of particle morphology parameters. The fact that the BC sample sets COJ300 and R2500U from Zhang et al. (2020) exhibit similar primary particle diameters and similar mobility diameters but differing fractal parameters allow us to isolate the influence of the configuration of the primary particles within the aggregates on their side/back scattering properties, holding other factors constant to the greatest possible extent.

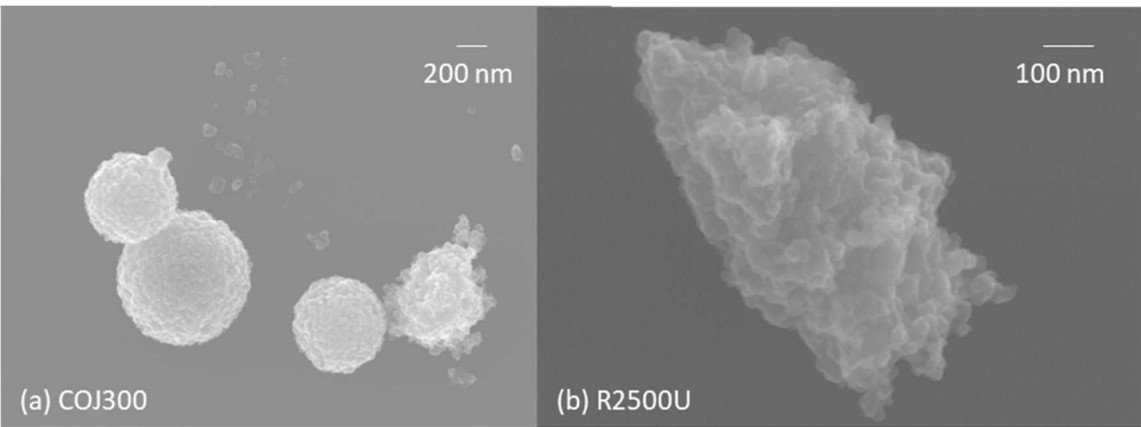

**Figure 1:** SEM images of size-selected 400-nm aggregates from sample set (a) COJ300 and (b) R2500U.

In the present study, we examine the scattering measurements at an angular range of 135±20° obtained with the SPectrometer for Ice Nuclei (SPIN; Garimella et al., 2016) instrumentation at 670-nm wavelength for the aforementioned two sets of samples, COJ300 and R2500U size-selected at 400 nm, from Zhang et al. (2020). Then, we conduct new theoretical calculations for comparison to the measured scattering in a similar manner to Sela and Haspel (2021), where the aggregates in each set consist of the same number of primary particles, $N_{pp}$, of the same primary particle size ($d_{pp}$) but differing configurations of the primary particles, and now focusing on BC aggregates. Thus, we can examine whether the side/back scattering tendencies found in Sela and Haspel (2021) are reproduced in actual measurements and how the configuration of the primary particles influences these tendencies.

In addition, given that the SPIN measurements are in situ measurements of scattering by individual particles rather than bulk scattering measurements, we have a unique opportunity to examine how the present set of measurements and calculations compare with previous measurements and calculations of side/back scattering by bare/uncoated BC aggregates, such as those presented in Bohren and Kho (1985), Lu and Sorensen (1994), Gustafson and Kolokolova (1999), Liu and Mishchenko (2005), Liu and Mishchenko (2007), Liu et al. (2008), Burton et al. (2013; 2014), Kahnert and Kanngießer (2020) and references therein, and Romshoo et al. (2021).

## 2 Methods

### 2.1 SPIN scattering measurements

Optical measurements were performed using a linear depolarization optical particle counter (OPC) associated with the SPIN instrument (Garimella et al., 2016). The SPIN OPC is equipped with a continuous-wave 500-mW 670-nm wavelength laser (Osela ILS-640-250-FTH-1.5MM-100uM). Particle measurements are made with four optical detectors. See Garimella et al. (2016) for a more complete description of the OPC geometry, including a diagram of the instrumentation. Size is measured based on side scattering with a detector situated at a zenith angle of 90˚ (i.e., 90˚ from the direction of propagation of the

incident laser beam) using a Mangin mirror pair. Three backscattering detectors measure the scattered photon counts according to polarization. The incident radiation from the laser is polarized with its electric field vector parallel to the scattering plane. Detectors P1 and P2 measure scattered photons with parallel polarization (the same polarization as the incident radiation), while detector S1 measures scattered photons with perpendicular polarization (electric field vectors perpendicular to the plane of the scattering). As mentioned in Sect. 1, these three detectors are each situated at a scattering zenith angle, $\theta_{sca}$, of 135° with a half angle of acceptance of 20°. Detectors P1 and S1 collect photons from the same scattered photon stream after it passes through a 50/50 polarizing beam splitter, while detector P2 collects photons from a separate photon stream that propagates at a different azimuthal angle with respect to the direction of propagation of the incident laser beam (but still propagates at a scattering zenith angle of 135±20°). For each of these two photon streams, the scattered laser light propagating at $\theta_{sca}$ = 135±20° first passes through a collimating lens, which transforms the scattered rays into parallel rays, followed by a focusing lens, which focuses the rays towards the detector. This lens configuration is intended to provide equal weight to each ray in the range $\theta_{sca}$ = 135±20° and approximately unit transmission. Scattering data for each particle is recorded in units of photon counts (photons per second). Given that the incident radiation from the laser is polarized parallel to the scattering plane, a higher photon count registered in detector S1 (in an absolute sense and/or relative to the photon counts registered in detectors P1 and P2) indicates some asymmetry/nonsphericity in the shape of the scattering particles or possibly birefringence or chirality in the scattering particle material. See Droplet Measurement Technologies, Inc. (2013) and Garimella et al. (2016) for more details on the SPIN instrumentation.

Particle generation and characterization of the BC samples followed the methodology outlined in Zhang et al. (2020). The size distributions of measured BC particles follow a Poisson/log-normal distribution. To avoid the influence of multiply charged BC particles, which could reach up to 16% of the total BC population, size thresholds corresponding to the 90% quantile of optical diameter (1310.7 nm for COJ300 and 6769.4 nm for R2500J, respectively) were applied to the particle-by-particle data. This filter accounts for the differences between the optical and mobility diameter while minimizing the impact of doubly and triply charged particles in our data analysis.

The relative humidity (RH) conditions of the SPIN experiments (62% at −50°C to 68% at −40°C) were below liquid water saturation. If any water vapor molecules had condensed onto the surfaces of the particles, they would have frozen immediately, resulting in an observable ice crystal signal. Ice crystals were not observed, and we therefore assume that the BC particles examined were dry.

## 2.2 Theoretical calculations

The theoretical aggregates are based on the mean microphysical properties of the COJ300 and R2500U 400 nm samples from Zhang et al. (2020), as listed in Sect. 1, but we also test the sensitivity of the results to variations in the overall aggregate diameter and to variations in $d_{pp}$. For each set of aggregates, first an SC aggregate with a roughly spherical outer

envelope is constructed, where in each SC aggregate, the primary particles touch but do not overlap (point contact). Our default SC aggregate has an outer-envelope diameter ($D_{\text{outer-envelope}}$) of 400 nm and a primary particle diameter of 35 nm.

Next, the IAS model of Stachurski (2003; 2011; 2013) is employed to construct a disordered but very compact and still roughly spherical aggregate with the same values of $D_{\text{outer-envelope}}$, $d_{\text{pp}}$, and $N_{\text{pp}}$ as the respective SC aggregate. (Refer to the description of the pairs of aggregates in Sela and Haspel (2021).) As with the SC aggregates, in the IAS aggregates, the primary particles touch one another but do not overlap, and each aggregate is monodisperse with respect to its primary particles.

Next, the fractal aggregate generating code of Mackowski (1995; 2006) is employed to generate two more aggregates based on a sequential cluster-cluster aggregation (CCA) algorithm. One of these two fractal aggregates is more compact in order to mimic the COJ300 samples, while the second of these two fractal aggregates is more extended in order to mimic the R2500U samples. Once again, as with the SC and IAS aggregates, in the CCA aggregates, the primary particles may touch but do not overlap, and each aggregate is monodisperse with respect to its primary particles. The CCA aggregates have the same values

of $d_{\text{pp}}$ and $N_{\text{pp}}$ as the SC and IAS aggregates but do *not* have the same outer envelope diameter (which in any case is not a meaningful diameter for such particles; see DeCarlo et al. (2004)). The more compact CCA aggregates have a significantly larger outer envelope and a significantly higher porosity than the SC and IAS aggregates. The more extended CCA aggregates have an outer envelope diameter that is even larger and a porosity that is even higher, as well as a more fractal appearance. (See, e.g., Figs. 2 and 3, where the explanations of the fractal parameters cited in the captions of Figs. 2 and 3

are contained in the following paragraphs.)

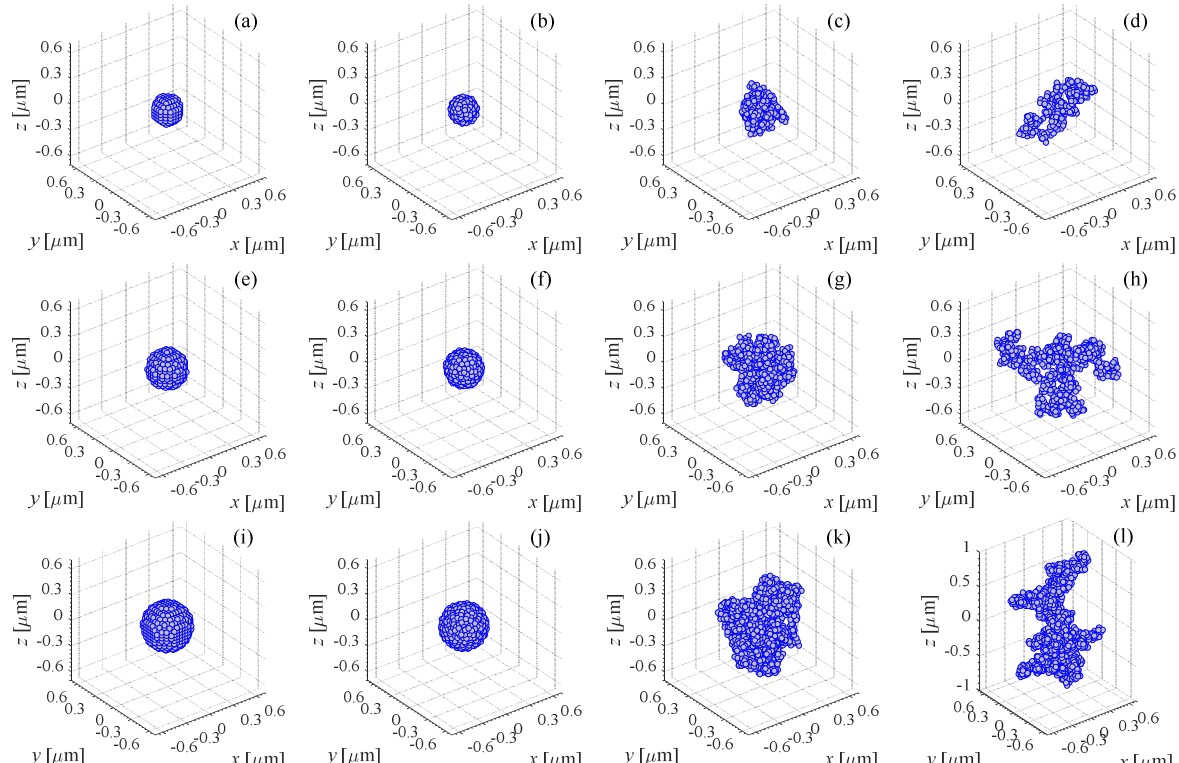

**Figure 2:** Positions of the primary particles for aggregates with $d_{pp}$ = 35 nm and varying values of $D_{\text{outer-envelope}}$ of the SC aggregate. Row 1 (aggregates with $D_{\text{outer-envelope}}$ of the SC aggregate = 300 nm, $N_{pp}$ = 317, $d_{pp}$ = 35 nm): (a) SC, (b) IAS, (c) CCA, $D_f$ = 2.34, $k_{\text{Sorensen}}$ = 1.085, (d) CCA, $D_f$ = 1.92, $k_{\text{Sorensen}}$ = 1.873. Row 2 (aggregates with $D_{\text{outer-envelope}}$ of the SC aggregate = 400 nm, $N_{pp}$ = 771, $d_{pp}$ = 35 nm): (e) SC, (f) IAS, (g) CCA, $D_f$ = 2.34, $k_{\text{Sorensen}}$ = 1.085, (h) CCA, $D_f$ = 1.92, $k_{\text{Sorensen}}$ = 1.873. Row 3 (aggregates with $D_{\text{outer-envelope}}$ of the SC aggregate = 500 nm, $N_{pp}$ = 1529, $d_{pp}$ = 35 nm): (i) SC, (j) IAS, (k) CCA, $D_f$ = 2.34, $k_{\text{Sorensen}}$ = 1.085, (l) CCA, $D_f$ = 1.92, $k_{\text{Sorensen}}$ = 1.873. Note the different scale of the $z$ -axis in plot (l).

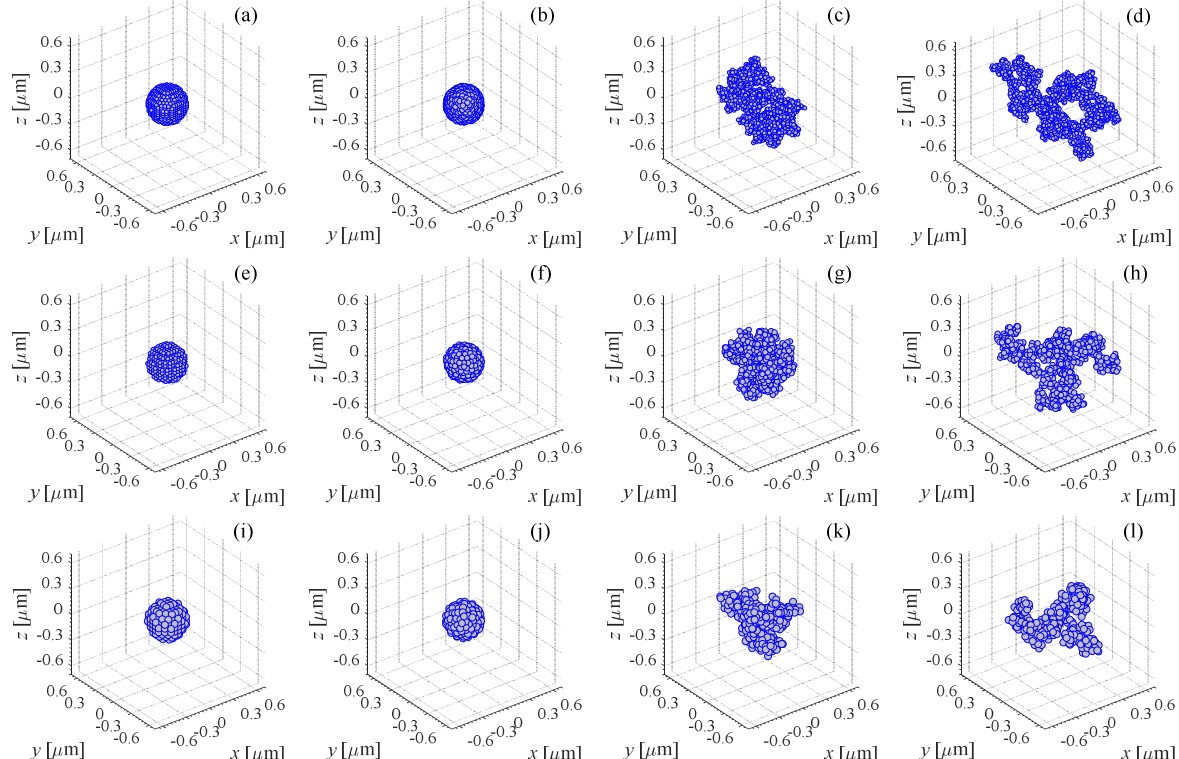

**Figure 3:** Positions of the primary particles for aggregates starting from an SC aggregate with $D_{\text{outer-envelope}}$ = 400 nm and varying values of $d_{\text{pp}}$. Row 1 (aggregates with $N_{\text{pp}}$ = 2106, $d_{\text{pp}}$ = 25 nm): (a) SC, (b) IAS, (c) CCA, $D_{\text{f}}$ = 2.34, $k_{\text{Sorensen}}$ = 1.085, (d) CCA, $D_{\text{f}}$ = 1.92, $k_{\text{Sorensen}}$ = 1.873. Row 2 (aggregates with $N_{\text{pp}}$ = 771, $d_{\text{pp}}$ = 35 nm): (e) SC, (f) IAS, (g) CCA, $D_{\text{f}}$ = 2.34, $k_{\text{Sorensen}}$ = 1.085, (h) CCA, $D_{\text{f}}$ = 1.92, $k_{\text{Sorensen}}$ = 1.873. Row 3 (aggregates with $N_{\text{pp}}$ = 377, $d_{\text{pp}}$ = 45 nm): (i) SC, (j) IAS, (k) CCA, $D_{\text{f}}$ = 2.34, $k_{\text{Sorensen}}$ = 1.085, (l) CCA, $D_{\text{f}}$ = 1.92, $k_{\text{Sorensen}}$ = 1.873. Note the different scale of the $x$-axis in plot (d).

The input to the fractal aggregate generating code of Mackowski (1995; 2006) consists of the value of $N_{\text{pp}}$, the radius of the primary particle, $a_{\text{pp}} = \frac{1}{2} d_{\text{pp}}$, the 3D fractal dimension, $D_{\text{f}}$, and the fractal pre-factor, labeled here $k_{\text{Sorensen}}$, from the following relationship (Sorensen, 2001):

$$N_{\text{pp}} = k_{\text{Sorensen}} \left( \frac{R_{\text{g}}}{a_{\text{pp}}} \right)^{D_{\text{f}}}, \tag{1}$$

where $R_g$ is the radius of gyration. As mentioned in Sect. 1, for COJ300, $D_f = 2.34$ with a 95% confidence interval range of 2.12-2.56, and for R2500U, $D_f = 1.92$ with a 95% confidence interval range of 1.68-2.16 (Zhang et al., 2020). Regarding the fractal pre-factor, by assuming that $R_g = \frac{1}{3}L_{max}$, where $L_{max}$ is the length of longest dimension of the aggregate periphery, Zhang et al. (2020) wrote a similar relationship to that of Sorensen (2001):

$$N_{pp} = k_{\text{Zhang et al.}} \left( \frac{L_{max}}{d_{pp}} \right)^{D_f} , \qquad (2)$$

where the fractal pre-factor as defined by Zhang et al. (2020) is labeled $k_{\text{Zhang et al.}}$. From this, we can obtain the following relationship between $k_{\text{Sorensen}}$ and $k_{\text{Zhang et al.}}$:

$$k_{\text{Sorensen}} = k_{\text{Zhang et al.}} \times \left( \frac{2}{3} \right)^{-D_f} . \qquad (3)$$

Based on the data from Zhang et al. (2020), e.g., as shown in their Fig. A6(c), for COJ300, $k_{\text{Zhang et al.}} = 0.42$, and for R2500U, $k_{\text{Zhang et al.}} = 0.86$. By using Eq. (3), together with the respective values of $D_f$ above, for COJ300, we obtain $k_{\text{Sorensen}} = 1.085$, and for R2500U, we obtain $k_{\text{Sorensen}} = 1.873$. Thus, the CCA aggregates with $D_f = 2.34$ and $k_{\text{Sorensen}} = 1.085$ appear more compact and mimic the COJ300 samples, while the CCA aggregates with $D_f = 1.92$ and $k_{\text{Sorensen}} = 1.873$ appear more extended and mimic the R2500U samples (refer again to Figs. 2 and 3). We also test different realizations of these CCA aggregates, varying the values of $D_f$ within the 95% confidence interval ranges stated above, as well as different realizations of the IAS aggregates.

We note that the fractal aggregate generating code of Mackowski (1995; 2006) includes an option to generate aggregates based on diffusion-limited particle-cluster aggregation (PCA). However, as discussed in Mackowski (1995; 2006) and in Filippov et al. (2000), for given values of $D_f$ and $k_{\text{Sorensen}}$, with the sequential CCA algorithm, Eq. (1) above is fulfilled exactly at each step. Thus, the sequential CCA algorithm should generate more precise fractal aggregates. From preliminary tests (not shown here), we find that on the whole, scattering calculations on aggregates generated using the sequential CCA option better reproduce some of the tendencies in the measured results than scattering calculations on aggregates generated using the PCA option. Thus, with respect to the fractal aggregates, by default, we present calculations for aggregates generated using the sequential CCA option of the fractal aggregate generating code of Mackowski (1995; 2006). However, when we vary the value of $D_f$ to its highest value within the 95% confidence interval range of $D_f$ for the COJ300 samples (2.56), the sequential CCA algorithm gives repeated error messages of "clusters did not combine" and produces a list of

primary particle positions that partially overlap one another. Therefore, for this highest value of $D_f$ only, we employ the PCA option of the fractal aggregate generating code of Mackowski (1995; 2006).

Even though the SC and IAS aggregates are not expected to represent either of the Zhang et al. (2020) sample sets well, these two configurations are useful to test for two reasons: (1) By constructing the SC and IAS aggregates of a given aggregate set first, we can determine how many primary particles of a given value of $d_{pp}$ fit compactly into sphere of a given value of $D_{outer-envelope}$. Then, as explained above, we use this same number of primary particles $N_{pp}$ with the same $d_{pp}$ to construct the fractal aggregates of the same set. By doing so, all of the aggregates of a given set possess the same

mass equivalent diameter, $D_{me}$, but varying configurations of the primary particles, which allows us to isolate the effect of the configuration of the primary particles, holding all other parameters constant to the greatest possible extent. (2) Although the SC and IAS aggregates are the most spherical of each set, they have a roughness on the nanometer scale and are not perfectly symmetric. Thus, even the SC and IAS aggregates should provide a minimal perpendicularly polarized scattered intensity against which the perpendicularly polarized scattered intensity provided by the fractal aggregates can be compared.

In testing the sensitivity of the results to variations in $D_{outer-envelope}$, we hold $d_{pp}$ constant at 35 nm and change the value of $D_{outer-envelope}$ of the SC aggregate to 300 nm or 500 nm. In testing the sensitivity of the results to variations in $d_{pp}$, we hold the value of $D_{outer-envelope}$ of the SC aggregate constant at 400 nm and change the value of $d_{pp}$ to 25 nm or 45 nm. Throughout the sensitivity studies, each individual aggregate is monodisperse *with respect to its primary particles*.

Once the aggregates are generated, we employ the MSTM model (Mackowski and Mishchenko, 1996) to calculate the

extinction efficiency, $Q_{ext\ MSTM}$, the absorption efficiency, $Q_{abs\ MSTM}$, the scattering efficiency, $Q_{sca\ MSTM}$, and the 4×4 scattering phase matrix, $\mathbf{S}$, of the aggregate at the wavelength of measurement, 670 nm. The default value of the complex refractive index of BC at 670 nm is taken to be $2.0+1.0i$ (Janzen, 1979; soot G of Fuller et al., 1999; Liu and Mishchenko, 2005, 2007; Liu et al., 2008; Moteki et al., 2010), where the real part, $m_{real}$, represents the refractive capability of the material, and the imaginary part, $m_{imag}$, represents the absorptive capability of the material, but the sensitivity to this choice

is also investigated.

The random-orientation option of MSTM (Mackowski, 2013) is used as a proxy for averaging over many different realizations of each of the IAS and fractal aggregates (see, e.g., Mishchenko et al. (2007) for an explanation of this), but we also test the sensitivity of the results to the choice of realization, and we also conduct simulations with fixed orientation.

The intensity of parallel polarized scattered radiation for parallel polarized incident radiation, $I_{sca_{\parallel \rightarrow \parallel}}$, is obtained from the

elements of the scattering phase matrix outputted from MSTM as:

$$I_{\text{sca}_{\parallel \to \parallel}}(\theta_{\text{sca}}) = \frac{1}{2}\frac{k^2 \sigma_{\text{sca}}}{4\pi}\left[\left(S_{11}(\theta_{\text{sca}}) + S_{12}(\theta_{\text{sca}})\right) + \left(S_{21}(\theta_{\text{sca}}) + S_{22}(\theta_{\text{sca}})\right)\right], \tag{4}$$

where $k = 2\pi/\lambda$ is the wave number, $\lambda$ is the wavelength, and $\sigma_{\text{sca}}$ is the scattering cross section of the aggregate with respect to unpolarized incident radiation (see also Sect. 3.2 regarding the scattering cross section). Similarly, the intensity of perpendicularly polarized scattered radiation for parallel polarized incident radiation, $I_{\text{sca}_{\parallel \to \perp}}$, is obtained from the elements of $\mathbf{S}$, as:

$$I_{\text{sca}_{\parallel \to \perp}}(\theta_{\text{sca}}) = \frac{1}{2}\frac{k^2 \sigma_{\text{sca}}}{4\pi}\left[\left(S_{11}(\theta_{\text{sca}}) + S_{12}(\theta_{\text{sca}})\right) - \left(S_{21}(\theta_{\text{sca}}) + S_{22}(\theta_{\text{sca}})\right)\right], \tag{5}$$

and the total intensity of scattered radiation as a function of scattering angle is given by the sum,

$$I_{\text{sca tot}}(\theta_{\text{sca}}) = I_{\text{sca}_{\parallel \to \parallel}}(\theta_{\text{sca}}) + I_{\text{sca}_{\parallel \to \perp}}(\theta_{\text{sca}}). \tag{6}$$

In using MSTM and Eqs. (4)-(6), we implicitly assume that the incident laser light is a 100% coherent plane wave that is 100% polarized parallel to the scattering plane. The scattered intensity over the range $\theta_{\text{sca}} = 135\pm20°$ is calculated as:

$$\begin{aligned}
I_{\text{sca}_{\parallel \to \parallel}}(135 \pm 20°) &= \int_{115°}^{155°} d\mu_{\text{sca}}\, I_{\text{sca}_{\parallel \to \parallel}}(\theta_{\text{sca}}) \\
I_{\text{sca}_{\parallel \to \perp}}(135 \pm 20°) &= \int_{115°}^{155°} d\mu_{\text{sca}}\, I_{\text{sca}_{\parallel \to \perp}}(\theta_{\text{sca}}), \\
I_{\text{sca tot}}(135 \pm 20°) &= \int_{115°}^{155°} d\mu_{\text{sca}}\, I_{\text{sca tot}}(\theta_{\text{sca}})
\end{aligned} \tag{7}$$

where $\mu_{\text{sca}} \equiv \cos\theta_{\text{sca}}$. In implementing Eq. (7) numerically, for each discrete value of scattering zenith angle, $\theta_{\text{sca}_i}$, $d\mu_{\text{sca}}$ is calculated explicitly as $\left|\cos\left(\theta_{\text{sca}_i} - 0.5°\right) - \cos\left(\theta_{\text{sca}_i} + 0.5°\right)\right|$, i.e., with a span of 1°. Note that in Eqs. (4)-(5), a factor of $\varepsilon c E_0^2/2$, where $\varepsilon$ is the electric permittivity of the background material (here assumed to be vacuum), $c$ is the speed of light in vacuum, and $E_0$ is the amplitude of the electric field of the incident electromagnetic wave (here assumed to be of unit value), which would give the expressions the true dimensionality of radiative intensity, has been suppressed. See Appendix A for a summary of the terms, acronyms, and symbols used in this study.

**3 Results**

**3.1 SPIN measurements**

A summary of the scattering measurements from the SPIN OPC is given in Table 1. The measurements are presented for the 5th, 25th, 50th, 75th, and 95th percentiles in order to demonstrate the scope of the variability and uncertainty in the data.

| | COJ300 | | | | | R2500U | | | | |
|---|---|---|---|---|---|---|---|---|---|---|
| Percentiles | 5 | 25 | 50 | 75 | 95 | 5 | 25 | 50 | 75 | 95 |
| P1 [photons s$^{-1}$] | 205.0 | 527.0 | 716.0 | 924.0 | 1318.0 | 126.6 | 567.0 | 1187.0 | 2327.8 | 5538.8 |
| P2 [photons s$^{-1}$] | 167.0 | 436.0 | 612.0 | 808.0 | 1163.0 | 80.0 | 418.8 | 964.0 | 1866.0 | 4356.8 |
| P=(P1+P2)/2 [photons s$^{-1}$] | 237.5 | 506.0 | 676.0 | 849.0 | 1159.0 | 170.1 | 569.6 | 1147.0 | 2147.4 | 4675.9 |
| S=S1 [photons s$^{-1}$] | 0.0 | 0.0 | 0.0 | 33.0 | 118.0 | 0.0 | 26.0 | 231.0 | 767.2 | 2408.9 |
| P+S [photons s$^{-1}$] | 252.0 | 523.0 | 693.0 | 876.0 | 1218.0 | 201.8 | 672.4 | 1462.0 | 2946.4 | 6848.0 |
| S/P* | 0.000 | 0.000 | 0.000 | 0.0390 | 0.102 | 0.000 | 0.046 | 0.201 | 0.357 | 0.515 |

*Obtained by dividing the "S" percentiles respectively by the corresponding "P" percentiles.

**Table 1.** Percentiles of filtered photon counts from the SPIN instrumentation. The percentiles of photon counts are given to a precision of a tenth of a photon s$^{-1}$, while the ratio S/P is listed to three significant figures.

From Table 1, the photon counts for the more fractal sample set, R2500U, are significantly higher and exhibit more variation than the photon counts for the more spherical sample set, COJ300. This is true both for each polarization individually and for the total P+S. Thus, from the SPIN measurements, we find a stronger scattering signal at scattering angles 135±20° from the more fractal sample set. We note that the P1 photon counts are higher than the P2 photon counts. However, this possibly systematic difference is small compared with the variability in photon counts from particle to particle.

Likewise, from Table 1, we see that the S/P ratio (the linear depolarization ratio) for R2500U is significantly higher than the S/P ratio for COJ300. For the R2500U sample set, the median value of S/P is 0.201, with 25th and 75th percentile values of 0.046 and 0.357, respectively, and a 95th percentile value greater than 0.5. In contrast, for the COJ300 sample set, the median value of S/P is 0.0, with 25th and 75th percentile values of 0.0 and 0.039, respectively. Overall, from our measurements, more than half (~60.4%) of the COJ300 particles have undetectably low S scattering signals and therefore

S/P values.

As mentioned in Sect. 2.1, a nonzero value of S when the incident radiation is polarized parallel to the scattering plane indicates some asymmetry/nonsphericity in the shape of the scattering particle (or possibly chirality or birefringence in the scattering particle material). Thus, the higher median S/P for the R2500U sample set corresponds with it being the more fractal sample set, exhibiting more irregular and extended shapes, while the zero median S/P ratio for the COJ300 sample set

corresponds well with it being the less fractal sample set, exhibiting shapes that are closer to spherically symmetric. At the same time, the nonzero 75th and 95th percentile values of S/P for COJ300 indicate that some of the COJ300 particles are

non-spherical, albeit less so than the R2500U, which also corresponds with the fact that the mean fractal dimension of the COJ300 samples ($D_f$ = 2.34) is lower than 3.

Below, we examine whether these tendencies are also reproduced in our theoretical calculations.

**3.2 Theoretical calculations – sensitivity to $D_{\text{outer-envelope}}$ of the SC aggregate**

The results for the aggregates shown in Fig. 2 (i.e., for aggregates generated starting with an SC aggregate with $D_{\text{outer-envelope}}$ = 300 nm and $d_{\text{pp}}$ = 35 nm, for aggregates generated starting with an SC aggregate with $D_{\text{outer-envelope}}$ = 400 nm and $d_{\text{pp}}$ = 35 nm (our default set), and for aggregates generated starting with an SC aggregate with $D_{\text{outer-envelope}}$ = 500 nm and $d_{\text{pp}}$ = 35 nm) are given in Table B1, in Fig. 4, and in Table 2.

In columns 2, 4, and 6 of Table B1, the values of $Q_{\text{ext MSTM}}$, $Q_{\text{abs MSTM}}$, and $Q_{\text{sca MSTM}}$ of each aggregate as given by MSTM are presented. The efficiencies as given by MSTM are with respect to the volume-mean radius, $R_{\text{volume-mean}}$, which is the radius of a sphere that has the same ratio of volume to surface area. For a monodisperse aggregate:

$$R_{\text{volume-mean}} = \left( \sum_{i=1}^{N_{\text{pp}}} a_{\text{pp}}^3 \right)^{1/3} = \left( N_{\text{pp}} a_{\text{pp}}^3 \right)^{1/3} = N_{\text{pp}}^{1/3} a_{\text{pp}}, \tag{8}$$

such that the extinction, absorption, and scattering cross sections, respectively ($\sigma_{\text{ext}}$, $\sigma_{\text{abs}}$, $\sigma_{\text{sca}}$) are given by:

$$\sigma_{\text{ext/abs/sca}} = Q_{\text{ext/abs/sca MSTM}} \times \pi \left( N_{\text{pp}}^{2/3} a_{\text{pp}}^2 \right). \tag{9}$$

The cross sections calculated based on Eq. (9) are given in columns 3, 5, and 7 of Table B1.

From Table B1, we see that the SC and IAS aggregates tend to have higher extinction, absorption, and scattering cross sections than the fractal aggregates in the same set. This agrees with the findings of Liu and Mishchenko (2005), Liu et al. (2008), and Romshoo et al. (2021), who found that the extinction and scattering cross sections of soot aggregates increase as 270 the aggregates become more compact. (See also the review in Kahnert and Kanngießer (2020).) For aggregates generated starting with an SC aggregate with $D_{\text{outer-envelope}}$ = 400 nm and $d_{\text{pp}}$ = 35 nm, and for aggregates generated starting with an SC aggregate with $D_{\text{outer-envelope}}$ = 500 nm and $d_{\text{pp}}$ = 35 nm, the SC aggregate exhibits the highest extinction cross section of the set, which corresponds with the results in Sela and Haspel (2021). For aggregates generated starting with an SC aggregate with $D_{\text{outer-envelope}}$ = 300 nm and $d_{\text{pp}}$ = 35 nm, the IAS aggregate exhibits the highest extinction cross section of 275 the set.

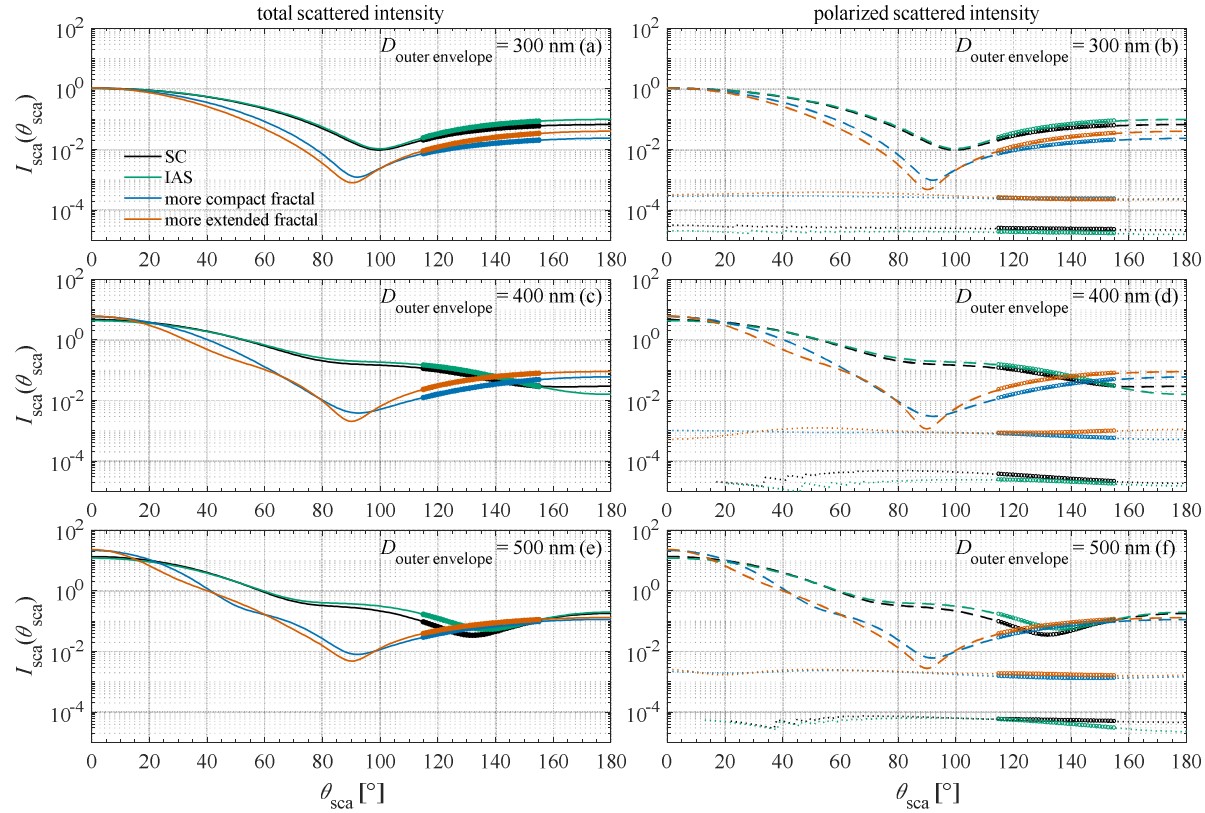

**Figure 4:** Scattered intensity as a function of scattering angle as obtained from MSTM for aggregates generated starting with an SC aggregate with $D_{\text{outer-envelope}}$ = 300 nm and $d_{\text{pp}}$ = 35 nm (plots a and b), for aggregates generated starting with an SC aggregate with $D_{\text{outer-envelope}}$ = 400 nm and $d_{\text{pp}}$ = 35 nm (our default set; plots c and d), and for aggregates generated starting with an SC aggregate with $D_{\text{outer-envelope}}$ = 500 nm and $d_{\text{pp}}$ = 35 nm (plots e and f). Plots (a), (c), and (e) contain the total scattered intensity (solid curves), while plots (b), (d), and (f) contain the scattered intensity polarized parallel (dashed curves) and perpendicular (dotted curves) to the scattering plane. The range $\theta_{\text{sca}}$ = 135±20° of the calculations is highlighted on each curve with a thicker curve.

In Figs. 4a and b, we show the scattered intensity as a function of scattering angle as obtained from MSTM for aggregates generated starting with an SC aggregate with $D_{\text{outer-envelope}}$ = 300 nm and $d_{\text{pp}}$ = 35 nm, where the range $\theta_{\text{sca}}$ = 135±20° is highlighted on each curve with a thicker curve. In Fig. 4a, the total scattered intensity is shown, while in Fig. 4b, the scattered intensity is separated according to polarization. From Fig. 4a, we see that the SC aggregate (black curve) exhibits a slightly higher scattered intensity in the direct forward direction ($\theta_{\text{sca}}$ = 0°) than the IAS aggregate (green curve), while the IAS aggregate exhibits a higher scattered intensity at scattering angles 135±20° than the SC aggregate, both of which agree with the results of Sela and Haspel (2021). In addition, the more extended fractal aggregate (CCA, $D_{\text{f}}$ = 1.92, $k_{\text{Sorensen}}$ =

1.873; red curve) exhibits a higher scattered intensity at scattering angles 135±20° than the more compact fractal aggregate (CCA, $D_f$ = 2.34, $k_{Sorensen}$ = 1.085; blue curve), which agrees with the SPIN measurements. However, the two fractal aggregates exhibit lower scattered intensities at scattering angles 135±20° than both the SC aggregate and the IAS aggregate, and this is due to the fact that the very compact SC and IAS aggregates exhibit higher scattering cross sections (refer to Table B1) and scatter more overall than the two fractal aggregates.

From Fig. 4b, we see that, as expected, all of the aggregates exhibit significantly more parallel polarized scattered intensity (the same polarization as the incident radiation; dashed curves) than perpendicularly polarized scattered intensity (dotted curves). Also as expected, we see that the SC and IAS aggregates exhibit a minimal but nonzero perpendicularly polarized scattered intensity (dotted black curve and dotted green curve, respectively); as mentioned in Sect. 2.2, these two aggregates are the most spherical of each set but contain a roughness on the nanometer scale and are not perfectly symmetric. (Refer also to Figs. 2 and 3.) From Fig. 4b, we also see that as with the total intensity, the more extended fractal aggregate (CCA, $D_f$ = 1.92, $k_{Sorensen}$ = 1.873; red dashed curve) exhibits a higher parallel polarized scattered intensity at scattering angles 135±20° than the more compact fractal aggregate (CCA, $D_f$ = 2.34, $k_{Sorensen}$ = 1.085; blue dashed curve), but at the same time, these two fractal aggregates exhibit lower parallel polarized scattered intensities at scattering angles 135±20° than the SC aggregate (black dashed curve) and the IAS aggregate (green dashed curve). Finally, from Fig. 4b, we see that at this size, the two fractal aggregates exhibit very similar perpendicularly polarized scattered intensities at scattering angles 135±20° to one another (dotted blue curve and dotted red curve, respectively).

In Figs. 4c and 4d, we show the scattered intensity as a function of scattering angle as obtained from MSTM for aggregates generated starting with an SC aggregate with $D_{outer-envelope}$ = 400 nm and $d_{pp}$ = 35 nm (our default set of aggregates). In Fig. 4c, the total scattered intensity is shown, while in Fig. 4d, the scattered intensity is separated according to polarization. We can see that the tendencies exhibited in Figs. 4c and 4d are similar to the tendencies exhibited in Figs. 4a and 4b, but with several distinctions. (1) There is more of a difference in the scattered intensity at scattering angles 135±20° between the more extended fractal aggregate and the more compact fractal aggregate, with the more extended fractal aggregate exhibiting a clearly higher total scattered intensity at scattering angles 135±20°, a clearly higher parallel polarized scattered intensity at scattering angles 135±20°, *and* a clearly higher perpendicularly polarized scattered intensity at scattering angles 135±20° than the more compact fractal aggregate, which agrees with the results from the SPIN measurements. In fact, we find that in this way, this default set of aggregates mimics the results from the SPIN measurements better than any set of aggregates that we tested. (2) In the range $\theta_{sca}$ = 135±20°, the curves of parallel polarized scattered intensity for the two fractal aggregates cross the curves of parallel polarized scattered intensity for the SC and IAS aggregates, which means that their values in that range are more comparable to those of the SC and IAS aggregates.

In Figs. 4e and 4f, we show the scattered intensity as a function of scattering angle as obtained from MSTM for aggregates generated starting with an SC aggregate with $D_{\text{outer-envelope}}$ = 500 nm and $d_{\text{pp}}$ = 35 nm. We can see that the tendencies exhibited in Figs. 4e and 4f are similar to the tendencies exhibited in Figs. 4a and 4b and in Figs. 4c and 4d, respectively. However, from Figs. 4e and 4f, we see that there is less of a difference in the scattered intensity at scattering angles 135±20° between the more extended fractal aggregate and the more compact fractal aggregate, as compared with the difference in scattered intensity at scattering angles 135±20° exhibited by the fractal aggregates in our default set of aggregates.

In Table 2, we list the values of scattered intensity over the range $\theta_{\text{sca}}$ = 135±20° corresponding to the curves in Fig. 4. From Table 2, we see the same tendencies as exhibited in Fig. 4 but now quantified. For example, looking at the values for aggregates generated starting with an SC aggregate with $D_{\text{outer-envelope}}$ = 400 nm and $d_{\text{pp}}$ = 35 nm (again, our default set of aggregates), the value of $I_{\text{sca}_{\parallel \to \parallel}}\left(135 \pm 20°\right)$ is higher for the more extended fractal aggregate ($2.423\times10^{-2}$ W m$^{-2}$) than for the more compact fractal aggregate ($1.344\times10^{-2}$ W m$^{-2}$). Likewise, the value of $I_{\text{sca}_{\parallel \to \perp}}\left(135 \pm 20°\right)$ is higher for the more extended fractal aggregate ($4.265\times10^{-4}$ W m$^{-2}$) than for the more compact fractal aggregate ($3.400\times10^{-4}$ W m$^{-2}$), and the value of $I_{\text{sca tot}}\left(135 \pm 20°\right)$ is higher for the more extended fractal aggregate ($2.465\times10^{-2}$ W m$^{-2}$) than for the more compact fractal aggregate ($1.378\times10^{-2}$ W m$^{-2}$). These tendencies resemble the tendencies from the SPIN measurements.

| Aggregate description | $I_{\text{sca}_{\parallel \to \parallel}}\left(135 \pm 20°\right)$ [W m$^{-2}$] | $I_{\text{sca}_{\parallel \to \perp}}\left(135 \pm 20°\right)$ [W m$^{-2}$] | Ratio of perpendicular to parallel $\dfrac{I_{\text{sca}_{\parallel \to \perp}}\left(135\pm20°\right)}{I_{\text{sca}_{\parallel \to \parallel}}\left(135\pm20°\right)}$ | $I_{\text{sca tot}}\left(135 \pm 20°\right)$ [W m$^{-2}$] |
|---|---|---|---|---|
| starting from an SC aggregate with $D_{\text{outer-envelope}}$ = 300 nm, $N_{\text{pp}}$ = 317, $d_{\text{pp}}$ = 35 nm | | | | |
| SC | $2.022\times10^{-2}$ | $1.174\times10^{-5}$ | $5.808\times10^{-4}$ | $2.023\times10^{-2}$ |
| IAS | $2.686\times10^{-2}$ | $8.778\times10^{-6}$ | $3.268\times10^{-4}$ | $2.687\times10^{-2}$ |
| More compact fractal: CCA, $D_{\text{f}}$ = 2.34, $k_{\text{Sorensen}}$ = 1.085 | $6.571\times10^{-3}$ | $1.159\times10^{-4}$ | $1.764\times10^{-2}$ | $6.687\times10^{-3}$ |
| More extended fractal: CCA, $D_{\text{f}}$ = 1.92, $k_{\text{Sorensen}}$ = 1.873 | $9.985\times10^{-3}$ | $1.139\times10^{-4}$ | $1.141\times10^{-2}$ | $1.010\times10^{-2}$ |
| starting from an SC aggregate with $D_{\text{outer-envelope}}$ = 400 nm, $N_{\text{pp}}$ = 771, $d_{\text{pp}}$ = 35 nm (our default set of aggregates) | | | | |
| SC | $3.425\times10^{-2}$ | $1.423\times10^{-5}$ | $4.155\times10^{-4}$ | $3.426\times10^{-2}$ |
| IAS | $4.444\times10^{-2}$ | $1.054\times10^{-5}$ | $2.371\times10^{-4}$ | $4.445\times10^{-2}$ |
| More compact fractal: CCA, $D_{\text{f}}$ = | $1.344\times10^{-2}$ | $3.400\times10^{-4}$ | $2.530\times10^{-2}$ | $1.378\times10^{-2}$ |

| | | | | |
|---|---|---|---|---|
| 2.34, $k_{\text{Sorensen}}$ = 1.085 | | | | |
| More extended fractal: CCA, $D_f$ = | | | | |
| 1.92, $k_{\text{Sorensen}}$ = 1.873 | 2.423×10$^{-2}$ | 4.265×10$^{-4}$ | 1.761×10$^{-2}$ | 2.465×10$^{-2}$ |
| starting from an SC aggregate with $D_{\text{outer-envelope}}$ = 500 nm, $N_{\text{pp}}$ = 1529, $d_{\text{pp}}$ = 35 nm | | | | |
| SC | 2.882×10$^{-2}$ | 2.656×10$^{-5}$ | 9.215×10$^{-4}$ | 2.885×10$^{-2}$ |
| IAS | 4.416×10$^{-2}$ | 2.199×10$^{-5}$ | 4.979×10$^{-4}$ | 4.419×10$^{-2}$ |
| More compact fractal: CCA, $D_f$ = | | | | |
| 2.34, $k_{\text{Sorensen}}$ = 1.085 | 2.938×10$^{-2}$ | 6.699×10$^{-4}$ | 2.280×10$^{-2}$ | 3.005×10$^{-2}$ |
| More extended fractal: CCA, $D_f$ = | | | | |
| 1.92, $k_{\text{Sorensen}}$ = 1.873 | 3.573×10$^{-2}$ | 8.581×10$^{-4}$ | 2.402×10$^{-2}$ | 3.659×10$^{-2}$ |

**Table 2.** Scattered intensity over the range $\theta_{\text{sca}}$ = 135±20° for aggregates generated starting with an SC aggregate with $D_{\text{outer-envelope}}$ = 300 nm and $d_{\text{pp}}$ = 35 nm, for aggregates generated starting with an SC aggregate with $D_{\text{outer-envelope}}$ = 400 nm and $d_{\text{pp}}$ = 35 nm, and for aggregates generated starting with an SC aggregate with $D_{\text{outer-envelope}}$ = 500 nm and $d_{\text{pp}}$ = 35 nm.

However, referring to Table 1, the relative differences in scattered photon counts between the R2500U samples and the COJ300 samples are larger than the relative differences in scattered intensity between the more extended fractal aggregate and the more compact aggregate shown in Table 2. For example, from the values in Table 1, the ratio of the median value of P+S for R2500U to the median value of P+S for COJ300 is 2.11, whereas from the values in Table 2, the ratio of the value of $I_{\text{sca tot}}\left(135\pm20°\right)$ for the more extended fractal aggregate from our default set of aggregates to the value of $I_{\text{sca tot}}\left(135\pm20°\right)$ for the more compact fractal aggregate from our default set of aggregates is 1.79. In addition, referring to Table 1, the highest ratio of perpendicularly polarized scattered intensity to parallel polarized scattered intensity listed, i.e., the value of S/P corresponding to the 95th percentile, is 0.055 for the COJ300 sample set and is 0.375 for the R2500U sample set; both of these values of S/P are higher than the values of $\dfrac{I_{\text{sca}\parallel\to\perp}\left(135\pm20°\right)}{I_{\text{sca}\parallel\to\parallel}\left(135\pm20°\right)}$ for the fractal aggregates in Table 2, which range from 1.141×10$^{-2}$ to 2.530×10$^{-2}$. This indicates that there were some samples measured in the SPIN measurements, especially in the R2500U sample set, that exhibit higher linear depolarization ratios than the theoretical aggregates shown in Fig. 2.

### 3.3 Theoretical calculations – sensitivity to $d_{\text{pp}}$

Results for aggregates generated starting with an SC aggregate with our default value of $D_{\text{outer-envelope}} = 400$ nm but with a smaller primary particle diameter of $d_{\text{pp}} = 25$ nm, and results for aggregates generated starting with an SC aggregate with

our default value of $D_{\text{outer-envelope}} = 400$ nm but with a larger primary particle diameter of $d_{\text{pp}} = 45$ nm are shown in Table B2, in Fig. 5, and in Table 3. The tendencies shown in Table B2, in Figs. 5a and 5b, in Figs. 5e and 5f, and in Table 3 are similar to those for our default set of aggregates. (Note that the scattered intensity as a function of scattering angle for our default set of aggregates from Figs. 4c and 4d are repeated as Figs. 5c and 5d for ease of comparison.) However, with $d_{\text{pp}} =$ 25 nm and with $d_{\text{pp}} = 45$ nm, again, there is less of a difference in the scattered intensity at scattering angles $135\pm20°$

between the more extended fractal aggregates and the more compact fractal aggregates, as compared with the difference exhibited by the fractal aggregates from our default set of aggregates with $d_{\text{pp}} = 35$ nm. Thus, once again, we find that our default set of aggregates mimics the results from the SPIN measurements better than any set of aggregates that we tested.

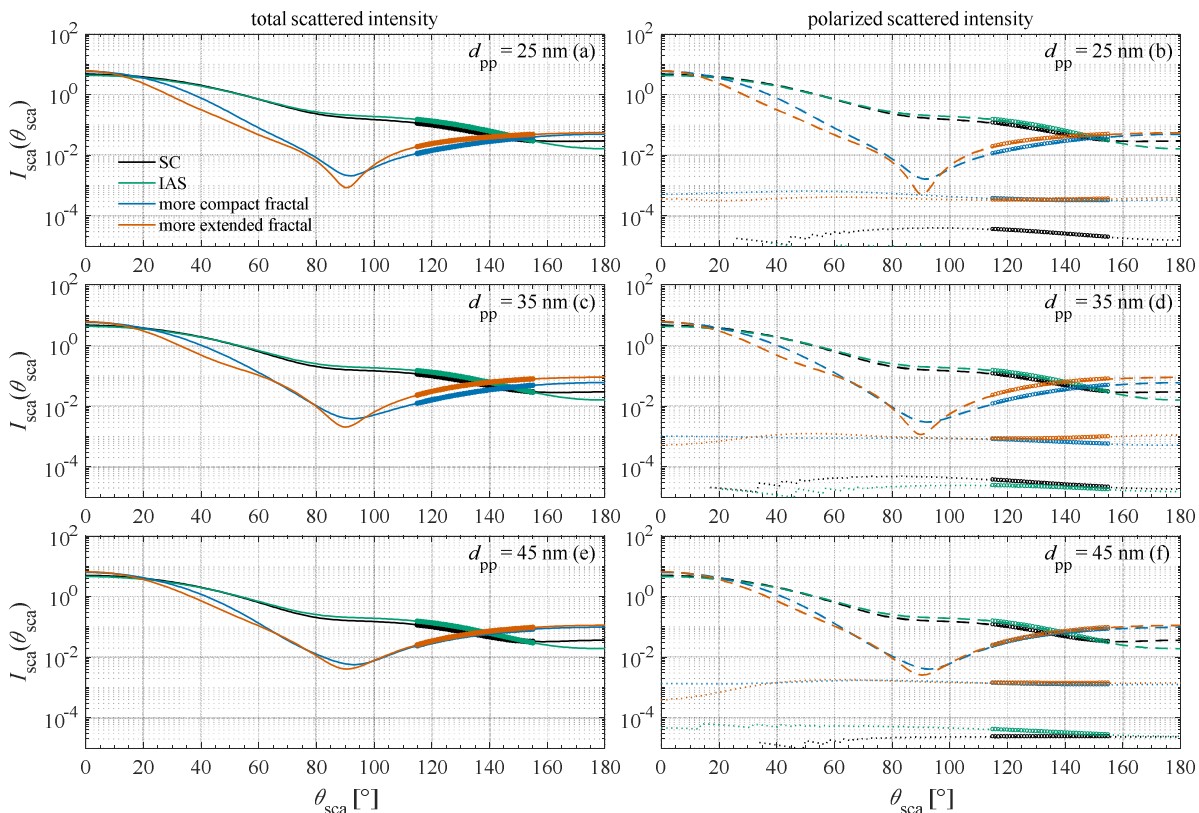


**Figure 5:** Scattered intensity as a function of scattering angle as obtained from MSTM for aggregates generated starting with an SC aggregate with $D_{\text{outer-envelope}} = 400$ nm and $d_{\text{pp}} = 25$ nm (plots a and b), for aggregates generated starting with an SC aggregate with

$D_{\text{outer-envelope}} = 400$ nm and $d_{pp} = 35$ nm (our default set; plots c and d), and for aggregates generated starting with an SC aggregate with $D_{\text{outer-envelope}} = 400$ nm and $d_{pp} = 45$ nm (plots e and f). Plots (a), (c), and (e) contain the total scattered intensity (solid curves),

while plots (b), (d), and (f) contain the scattered intensity polarized parallel (dashed curves) and perpendicular (dotted curves) to the scattering plane. The range $\theta_{\text{sca}} = 135\pm20°$ of the calculations is highlighted on each curve with a thicker curve.

| Aggregate description | $I_{\text{sca}_{\|\to\|}}(135\pm20°)$ [W m$^{-2}$] | $I_{\text{sca}_{\|\to\perp}}(135\pm20°)$ [W m$^{-2}$] | Ratio of perpendicular to parallel $\dfrac{I_{\text{sca}_{\|\to\perp}}(135\pm20°)}{I_{\text{sca}_{\|\to\|}}(135\pm20°)}$ | $I_{\text{sca tot}}(135\pm20°)$ [W m$^{-2}$] |
|---|---|---|---|---|
| starting from an SC aggregate with $D_{\text{outer-envelope}} = 400$ nm, $N_{pp} = 2106$, $d_{pp} = 25$ nm | | | | |
| SC | 3.418×10$^{-2}$ | 1.394×10$^{-5}$ | 4.077×10$^{-4}$ | 3.420×10$^{-2}$ |
| IAS | 4.513×10$^{-2}$ | 3.513×10$^{-6}$ | 7.785×10$^{-5}$ | 4.513×10$^{-2}$ |
| More compact fractal: CCA, $D_f = 2.34$, $k_{\text{Sorensen}} = 1.085$ | 1.142×10$^{-2}$ | 1.644×10$^{-4}$ | 1.440×10$^{-2}$ | 1.158×10$^{-2}$ |
| More extended fractal: CCA, $D_f = 1.92$, $k_{\text{Sorensen}} = 1.873$ | 1.662×10$^{-2}$ | 1.652×10$^{-4}$ | 9.939×10$^{-3}$ | 1.678×10$^{-2}$ |
| starting from an SC aggregate with $D_{\text{outer-envelope}} = 400$ nm, $N_{pp} = 377$, $d_{pp} = 45$ nm | | | | |
| SC | 3.443×10$^{-2}$ | 1.166×10$^{-5}$ | 3.386×10$^{-4}$ | 3.445×10$^{-2}$ |
| IAS | 4.542×10$^{-2}$ | 1.703×10$^{-5}$ | 3.750×10$^{-4}$ | 4.544×10$^{-2}$ |
| More compact fractal: CCA, $D_f = 2.34$, $k_{\text{Sorensen}} = 1.085$ | 2.537×10$^{-2}$ | 6.307×10$^{-4}$ | 2.486×10$^{-2}$ | 2.600×10$^{-2}$ |
| More extended fractal: CCA, $D_f = 1.92$, $k_{\text{Sorensen}} = 1.873$ | 2.774×10$^{-2}$ | 6.838×10$^{-4}$ | 2.465×10$^{-2}$ | 2.842×10$^{-2}$ |

**Table 3.** Scattered intensity over the range $\theta_{\text{sca}} = 135\pm20°$ for aggregates generated starting with an SC aggregate with $D_{\text{outer-envelope}} = 400$ nm and $d_{pp} = 25$ nm and 45 nm, respectively.

### 3.4 Theoretical calculations – sensitivity to complex refractive index

To test the sensitivity of our results to the assumed complex refractive index, we repeat the calculations on our default

aggregates (aggregates generated starting with an SC aggregate with $D_{\text{outer-envelope}} = 400$ nm and $d_{pp} = 35$ nm) with three

additional complex refractive indices that have been tabulated for non-graphitic light absorbing carbon: (1) $m = 1.75 + 0.63i$,

the lowest complex refractive index from Table 5 of Bond and Bergstrom (2006); (2) $m=1.85+0.71i$, the complex refractive index in the middle of the range from Table 5 of Bond and Bergstrom (2006) and that adopted by Bond et al. (2006); and (3) $m=2.26+1.26i$, the complex refractive index retrieved by Moteki et al. (2010). These complex refractive

indices were not necessarily tabulated at the identical wavelength of 670 nm, but they bracket a reasonable range of possible values of refractive indices of black carbon at wavelengths relevant to incident solar radiation (500-1064 nm) (Janzen, 1979; Fuller et al., 1999; Bond and Bergstrom, 2006; Bond et al., 2006; Liu and Mishchenko, 2005, 2007; Liu et al., 2008; Moteki et al., 2010). (See also the review in Kahnert and Kanngießer (2020).) For this sensitivity test, each of these three additional complex refractive indices in turn are set to be the complex refractive index of the primary particles in the aggregate. Results

for our default aggregates, but with primary particle complex refractive indices of $m=1.75+0.63i$, $m=1.85+0.71i$, $m=2.26+1.26i$, respectively, are shown in Table B3, in Fig. 6, and in Table 4.

From Table B3, we see that the higher the complex refractive index of the primary particles, the higher the extinction, absorption, and scattering cross sections of the aggregates, respectively, as would be expected. (See, also, Liu et al. (2008).) From Fig. 6, we see that the higher the complex refractive index, the farther towards the end of the $\theta_{sca}$ = 135±20° range the

curves of parallel polarized scattered intensity for the two fractal aggregates cross the curves of parallel polarized scattered intensity for the SC and IAS aggregates. From Table 4, we also see that for the fractal aggregates, the ratio of perpendicularly polarized to parallel polarized scattered radiation in the angular range $\theta_{sca}$ = 135±20° increases with the magnitude of the refractive index, which agrees with the findings of Bescond et al. (2013) regarding the direct backscatter depolarization caused by BC aggregates. Aside from that, the tendencies shown in Table B3, in Fig. 6, and in Table 4 are

quite similar to those for our default set of aggregates.

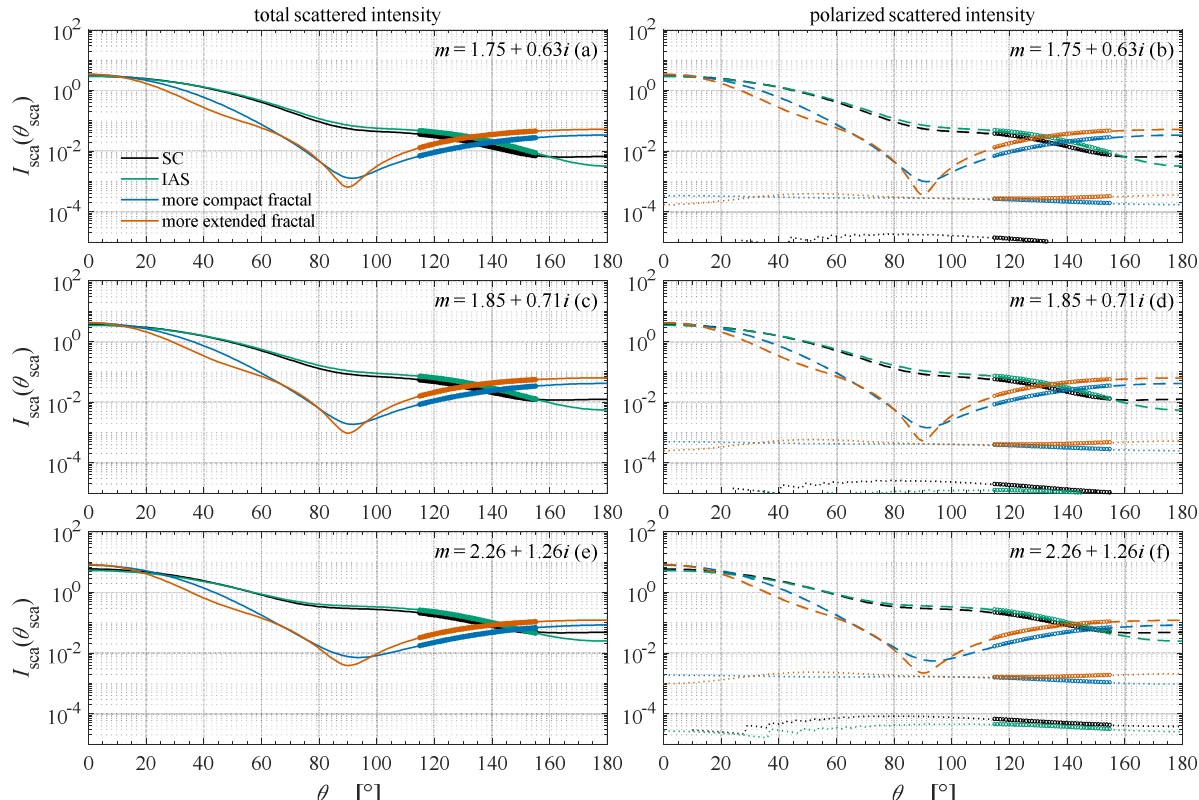

**Figure 6:** Scattered intensity as a function of scattering angle as obtained from MSTM for our default aggregates (aggregates generated starting with an SC aggregate with $D_{\text{outer-envelope}}$ = 400 nm and $d_{\text{pp}}$ = 35 nm), with primary particle refractive index $m=1.75+0.63i$ (plots a and b), with primary particle refractive index $m=1.85+0.71i$ (plots c and d), and with primary particle refractive index $m=2.26+1.26i$ (plots e and f). Plots (a), (c), and (e) contain the total scattered intensity (solid curves), while plots (b), (d), and (f) contain the scattered intensity polarized parallel (dashed curves) and perpendicular (dotted curves) to the scattering plane. The range $\theta_{\text{sca}}$ = $135\pm20°$ of the calculations is highlighted on each curve with a thicker curve.

| Aggregate description | $I_{\text{sca}_{\parallel\to\parallel}}$ $(135\pm20°$ | $I_{\text{sca}_{\parallel\to\perp}}$ $(135\pm20°$ | Ratio of perpendicular to parallel | $I_{\text{sca tot}}$ $(135\pm20$ |
|---|---|---|---|---|
| | [W m$^{-2}$] | [W m$^{-2}$] | $\dfrac{I_{\text{sca}_{\parallel\to\perp}}\left(135\pm20°\right)}{I_{\text{sca}_{\parallel\to\parallel}}\left(135\pm20°\right)}$ | [W m$^{-2}$] |
| starting from an SC aggregate with $D_{\text{outer-envelope}}$ = 400 nm, $N_{\text{pp}}$ = 771, $d_{\text{pp}}$ = 35 nm; $m=1.75+0.63i$ | | | | |
| SC | 1.054×10$^{-2}$ | 5.121×10$^{-6}$ | 4.861×10$^{-4}$ | 1.054×10$^{-2}$ |
| IAS | 1.469×10$^{-2}$ | 3.760×10$^{-6}$ | 2.560×10$^{-4}$ | 1.469×10$^{-2}$ |
| More compact fractal: CCA, $D_{\text{f}}$ = 2.34, $k_{\text{Sorensen}}$ = 1.085 | 7.719×10$^{-3}$ | 1.098×10$^{-4}$ | 1.422×10$^{-2}$ | 7.829×10$^{-3}$ |
| More extended fractal: CCA, | 1.403×10$^{-2}$ | 1.354×10$^{-4}$ | 9.652×10$^{-3}$ | 1.416×10$^{-2}$ |

| | | | | |
|---|---|---|---|---|
| $D_f$ = 1.92, $k_{Sorensen}$ = 1.873 | | | | |
| starting from an SC aggregate with $D_{outer\text{-}envelope}$ = 400 nm, $N_{pp}$ = 771, $d_{pp}$ = 35 nm; $m=1.85+0.71i$ | | | | |
| SC | $1.590\times10^{-2}$ | $7.240\times10^{-6}$ | $4.554\times10^{-4}$ | $1.590\times10^{-2}$ |
| IAS | $2.131\times10^{-2}$ | $5.424\times10^{-6}$ | $2.546\times10^{-4}$ | $2.131\times10^{-2}$ |
| More compact fractal: CCA, $D_f$ = 2.34, $k_{Sorensen}$ = 1.085 | $9.361\times10^{-3}$ | $1.612\times10^{-4}$ | $1.722\times10^{-2}$ | $9.522\times10^{-3}$ |
| More extended fractal: CCA, $D_f$ = 1.92, $k_{Sorensen}$ = 1.873 | $1.692\times10^{-2}$ | $1.991\times10^{-4}$ | $1.177\times10^{-2}$ | $1.712\times10^{-2}$ |
| starting from an SC aggregate with $D_{outer\text{-}envelope}$ = 400 nm, $N_{pp}$ = 771, $d_{pp}$ = 35 nm; $m=2.26+1.26i$ | | | | |
| SC | $6.122\times10^{-2}$ | $2.603\times10^{-5}$ | $4.252\times10^{-4}$ | $6.125\times10^{-2}$ |
| IAS | $7.623\times10^{-2}$ | $1.885\times10^{-5}$ | $2.473\times10^{-4}$ | $7.625\times10^{-2}$ |
| More compact fractal: CCA, $D_f$ = 2.34, $k_{Sorensen}$ = 1.085 | $1.863\times10^{-2}$ | $6.415\times10^{-4}$ | $3.443\times10^{-2}$ | $1.927\times10^{-2}$ |
| More extended fractal: CCA, $D_f$ = 1.92, $k_{Sorensen}$ = 1.873 | $3.302\times10^{-2}$ | $8.047\times10^{-4}$ | $2.437\times10^{-2}$ | $3.382\times10^{-2}$ |

**Table 4.** Scattered intensity over the range $\theta_{sca}$ = 135±20° for aggregates generated starting with an SC aggregate with $D_{outer\text{-}envelope}$ = 400 nm and $d_{pp}$ = 35 nm with different primary particle refractive indices.


### 3.5 Theoretical calculations – sensitivity to realizations of aggregate generating algorithms

Liu and Mishchenko (2007) found that varying the geometrical configuration of the primary particles in a soot cluster for fixed values of $D_f$, $k_{Sorensen}$, $N_{pp}$, and $d_{pp}$ has a weak effect on scattering and absorption in the visible part of the spectrum. Here we test the sensitivity to aggregate realization in a similar manner, but with respect to the Zhang et al. (2020)

experimental configuration and our associated theoretical aggregate parameters. We test additional realizations of the IAS aggregate of default size, the more compact fractal aggregate of default size, and the more extended fractal aggregate of default size, respectively, where the values of $N_{pp}$ (771) and $d_{pp}$ (35 nm) are identical for all of the realizations. First, we create two additional realizations of the IAS aggregate of default size, five additional realizations of the more compact fractal aggregate of default size with its default fractal parameters (CCA, $D_f$ = 2.34, $k_{Sorensen}$ = 1.085), and five additional

realizations of the more extended fractal aggregate of default size with its default fractal parameters (CCA, $D_f$ = 1.92,

$k_{Sorensen}$ = 1.873). Then we create six additional realizations of the more compact fractal aggregate of default size with the minimum value of $D_f$ within the 95% confidence interval range mentioned in Sects. 1 and 2.2 for the COJ300 samples (CCA, $D_f$ = 2.12, $k_{Sorensen}$ = 0.992), six additional realizations of the more compact fractal aggregate of default size with the maximum value of $D_f$ within the 95% confidence interval range for the COJ300 samples (PCA, $D_f$ = 2.56, $k_{Sorensen}$ =

1.186), six additional realizations of the more extended fractal aggregate of default size with the minimum value of $D_f$ within the 95% confidence interval range for the R2500U samples (CCA, $D_f$ = 1.68, $k_{Sorensen}$ = 1.700), and six additional realizations of the more extended fractal aggregate of default size with the maximum value of $D_f$ within the 95% confidence interval range for the R2500U samples (CCA, $D_f$ = 2.16, $k_{Sorensen}$ = 2.065). Note that in the deriving the 95% confidence interval range of $D_f$, the regression parameter $k_{Zhang\ et\ al.}$ was held constant. Accordingly, for each new value of $D_f$, a

new value of $k_{Sorensen}$ was calculated from Eq. (3) using the value of $k_{Zhang\ et\ al.}$ for the corresponding sample set. Note also that as stated in Sect. 2.2, when we vary the value of $D_f$ to its highest value within the 95% confidence interval range for the COJ300 samples, which is $D_f$ = 2.56, we employ the PCA option of the fractal aggregate generating algorithm rather than the CCA option, while for all of the other realizations, we employ the CCA option of the fractal aggregate generating algorithm.

In Table B4, we list the ranges of values of extinction, absorption, and scattering cross section for all of the realizations of the aggregate generating algorithms. From Table B4, we see again that the most compact aggregates (SC, IAS, PCA with $D_f$ = 2.56 and $k_{Sorensen}$ = 1.186, and CCA with $D_f$ = 2.16 and $k_{Sorensen}$ = 2.065) tend to have the highest extinction cross sections and scattering cross sections, which again agrees with the results of Liu and Mishchenko (2005), Liu et al. (2008), and Romshoo et al. (2021).

In Fig. 7, we show the scattered intensity as a function of scattering angle as obtained from MSTM for all of the realizations of the aggregate generating algorithms. From Figs. 7a and b, we see that there is hardly any discernable difference in the scattering patterns of the three IAS realizations (green curves), with just a small amount of discernable spread only in the very low values of perpendicularly polarized scattered intensity.

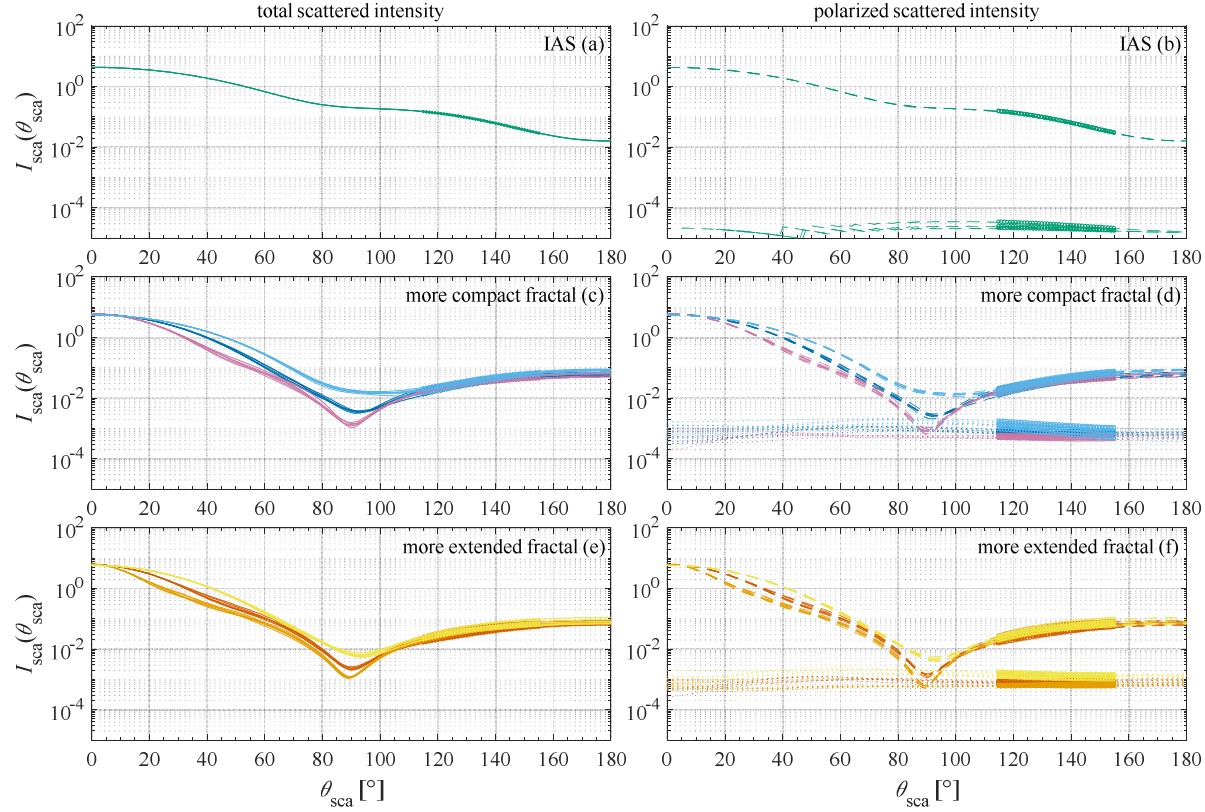


**Figure 7:** Scattered intensity as a function of scattering angle as obtained from MSTM for different realizations of the default IAS, default more compact fractal aggregate and default more extended fractal aggregate (all generated starting with an SC aggregate with $D_{\text{outer-envelope}}$ = 400 nm and $d_{\text{pp}}$ = 35 nm): (a),(b) different realizations of the default IAS aggregate (green curves); (c),(d) different realizations of the default more compact fractal aggregate (purple curves: $D_{\text{f}}$ = 2.12, $k_{\text{Sorensen}}$ = 0.992; blue curves: $D_{\text{f}}$ = 2.34,

$k_{\text{Sorensen}}$ = 1.085; light blue curves: PCA rather than CCA; $D_{\text{f}}$ = 2.56, $k_{\text{Sorensen}}$ = 1.186); (e),(f) different realizations of default more extended fractal aggregate (orange curves: $D_{\text{f}}$ = 1.68, $k_{\text{Sorensen}}$ = 1.700; red curves: $D_{\text{f}}$ = 1.92, $k_{\text{Sorensen}}$ = 1.873; yellow curves: $D_{\text{f}}$ = 2.16, $k_{\text{Sorensen}}$ = 2.065). Plots (a), (c), and (e) contain the total scattered intensity (solid curves), while plots (b), (d), and (f) contain the scattered intensity polarized parallel (dashed curves) and perpendicular (dotted curves) to the scattering plane. The range $\theta_{\text{sca}}$ = 135±20° of the calculations is highlighted on each curve with a thicker curve.


From Figs. 7c-f, where each color represents a group of realizations of fractal aggregates with identical values of $D_{\text{f}}$, $k_{\text{Sorensen}}$, $N_{\text{pp}}$, and $d_{\text{pp}}$, we see that as Liu and Mishchenko (2007) found, there is indeed a similarity to the scattering patterns of each group of curves. However, at the same time, there is some discernable spread in the scattering patterns, including in the range $\theta_{\text{sca}}$ = 135±20°. From Figs. 7c-f, we see that *within each graph*, the more compact the fractal

aggregate, the higher the values of scattered intensity over nearly the entire range of scattering angles, including over the range $\theta_{\text{sca}}$ = 135±20°. In Figs. 7c and 7d, which are different realizations of the more compact fractal aggregate (CCA, $D_{\text{f}}$ =

2.34, $k_{\text{Sorensen}}$ = 1.085), the light blue curves (PCA, $D_{\text{f}}$ = 2.56, $k_{\text{Sorensen}}$ = 1.186) lie largely above the blue curves (CCA, $D_{\text{f}}$ = 2.34, $k_{\text{Sorensen}}$ = 1.085), which in turn lie largely above the purple curves (CCA, $D_{\text{f}}$ = 2.12, $k_{\text{Sorensen}}$ = 0.992). In Figs. 7e and 7f, which are different realizations of the more extended fractal aggregate (CCA, $D_{\text{f}}$ = 1.92, $k_{\text{Sorensen}}$ = 1.873),

the yellow curves (CCA, $D_{\text{f}}$ = 2.16, $k_{\text{Sorensen}}$ = 2.065) lie largely above the red curves (CCA, $D_{\text{f}}$ = 1.92, $k_{\text{Sorensen}}$ = 1.873), which in turn lie largely above the orange curves (CCA, $D_{\text{f}}$ = 1.68, $k_{\text{Sorensen}}$ = 1.700). This is true of the total scattered intensity, as well as of the parallel polarized and perpendicularly polarized scattered intensities, and corresponds with the fact that the scattering cross sections of the more compact fractal aggregates as calculated from the output of MSTM are higher than the scattering cross sections of the more extended fractal aggregates. (Refer to Table B4.)

Only in the direct forward scattering direction does the scattered intensity of the more extended fractal aggregates in each graph increase above the scattered intensity of the more compact fractal aggregates, and this is only to a small extent that is difficult to discern by eye from the graphs. This is despite the fact that the extinction cross sections of the more extended aggregates are lower than the extinction cross sections of the more compact aggregates (again, refer to Table B4) and is probably due to the larger overall outer envelopes of the more extended aggregates (refer to Figs. 2 and 3). Due to the larger

overall outer envelopes of the more extended aggregates, their normalized scattering phase functions, exhibit stronger and narrower forward scattering peaks (see, e.g., Bohren and Kho (1985), Gustafson and Kolokolova (1999), and Liu and Mishchenko (2005), their Fig. 2). Even though the elements of the scattering phase matrix are multiplied by $\sigma_{\text{sca}}$ in converting from $\mathbf{S}$ to $I_{\text{sca}}(\theta_{\text{sca}})$ in Eqs. (4)-(5), the multiplication by $\sigma_{\text{sca}}$ is not enough to increase the directly forward scattered intensity in the broader forward scattering peak exhibited by the more compact aggregates to values greater than

the directly forward scattered intensity in the narrower forwarding scattering peak exhibited by the more extended aggregates.

In Table 5, we list the ranges of values of scattered intensity over the range $\theta_{\text{sca}}$ = 135±20° corresponding to Fig. 7. As was evident from Fig. 7, we can see from Table 5 that *within each category*, the more compact the fractal aggregate, the higher the values of scattered intensity in the angular range $\theta_{\text{sca}}$ = 135±20°. However, the overall range of perpendicularly

polarized scattered intensity in the angular range $\theta_{\text{sca}}$ = 135±20° for the more extended fractal aggregates ((2.883 − 7.706)×10⁻⁴) is higher than the overall range of perpendicularly polarized scattered intensity in the angular range $\theta_{\text{sca}}$ = 135±20° for the more compact fractal aggregates ((2.153 − 6.899)×10⁻⁴), and both of these ranges of intensity values encompass values that are more than an order of magnitude higher than the values of perpendicularly polarized scattered intensity for the SC aggregate (1.423×10⁻⁵) and for the IAS aggregates (9.524×10⁻⁶ − 1.391×10⁻⁵), all of which agrees with

the direction of the SPIN measurements. On the other hand, the highest ratio of perpendicularly polarized scattered intensity to parallel polarized scattered intensity in the angular range $\theta_{\text{sca}}$ = 135±20° among all of the fractal aggregates is

3.133×10$^{-2}$, which is lower than the 95th percentile value of the ratio S/P of either of the sample sets from the SPIN measurements (0.102 and 0.515, respectively; refer to Table 1).

| Aggregate description | Range of values of $I_{\mathrm{sca}_{\|\to\|}}(135\pm20°)$ [W m$^{-2}$] | Range of values of $I_{\mathrm{sca}_{\|\to\perp}}(135\pm20°)$ [W m$^{-2}$] | Range of values of ratio of perpendicular to parallel $\dfrac{I_{\mathrm{sca}_{\|\to\perp}}(135\pm20°)}{I_{\mathrm{sca}_{\|\to\|}}(135\pm20°)}$ | Range of values of $I_{\mathrm{sca\,tot}}(135\pm20°)$ [W m$^{-2}$] |
|---|---|---|---|---|
| SC | 3.425×10$^{-2}$ | 1.423×10$^{-5}$ | 4.155×10$^{-4}$ | 3.426×10$^{-2}$ |
| IAS | (4.421 − 4.506)×10$^{-2}$ | 9.524×10$^{-6}$ − 1.391×10$^{-5}$ | (2.154 − 3.087)×10$^{-4}$ | (4.422 − 4.507)×10$^{-2}$ |
| More compact fractal | | | | |
| CCA, $D_{\mathrm{f}}$ = 2.34, $k_{\mathrm{Sorensen}}$ = 1.085 | (1.344 − 1.907)×10$^{-2}$ | (3.050 − 4.047)×10$^{-4}$ | (1.751 − 2.530)×10$^{-2}$ | (1.378 − 1.940)×10$^{-2}$ |
| CCA, $D_{\mathrm{f}}$ = 2.12, $k_{\mathrm{Sorensen}}$ = 0.992 | (1.348 − 1.854)×10$^{-2}$ | (2.153 − 2.848)×10$^{-4}$ | (1.324 − 1.864)×10$^{-2}$ | (1.370 − 1.880)×10$^{-2}$ |
| PCA, $D_{\mathrm{f}}$ = 2.56, $k_{\mathrm{Sorensen}}$ = 1.186 | (1.484 − 2.202)×10$^{-2}$ | (3.371 − 6.899)×10$^{-4}$ | (2.271 − 3.133)×10$^{-2}$ | (1.518 − 2.271)×10$^{-2}$ |
| More compact fractal aggregates, overall | (1.344 − 2.202)×10$^{-2}$ | (2.153 − 6.899)×10$^{-4}$ | (1.324 − 3.133)×10$^{-2}$ | (1.370 − 2.271)×10$^{-2}$ |
| More extended fractal | | | | |
| CCA, $D_{\mathrm{f}}$ = 1.92, $k_{\mathrm{Sorensen}}$ = 1.873 | (1.480 − 2.695)×10$^{-2}$ | (3.047 − 4.314)×10$^{-4}$ | (1.427 − 2.213)×10$^{-2}$ | (1.512 − 2.733)×10$^{-2}$ |
| CCA, $D_{\mathrm{f}}$ = 1.68, $k_{\mathrm{Sorensen}}$ = 1.700 | (1.870 − 2.163)×10$^{-2}$ | (2.883 − 3.301)×10$^{-4}$ | (1.435 − 1.681)×10$^{-2}$ | (1.899 − 2.194)×10$^{-2}$ |
| CCA, $D_{\mathrm{f}}$ = 2.16, $k_{\mathrm{Sorensen}}$ = 2.065 | (1.871 − 2.894)×10$^{-2}$ | (5.176 − 7.706)×10$^{-4}$ | (2.321 − 3.089)×10$^{-2}$ | (1.923 − 2.971)×10$^{-2}$ |
| More extended fractal aggregates, overall | (1.480 − 2.894)×10$^{-2}$ | (2.883 − 7.706)×10$^{-4}$ | (1.427 − 3.089)×10$^{-2}$ | (1.512 − 2.971)×10$^{-2}$ |


**Table 5.** Ranges of values of scattered intensity over the range $\theta_{\mathrm{sca}}$ = 135±20° for different realizations of aggregates generated starting with an SC aggregate with $D_{\mathrm{outer\text{-}envelope}}$ = 400 nm and $d_{\mathrm{pp}}$ = 35 nm. The values for the single SC realization from Table 2 are also included here for reference.

In Sect. 3.6, we explore the range of values obtained with these same aggregate realizations, but with fixed aggregate orientation in the MSTM model calculations.

### 3.6 Theoretical calculations – fixed orientation versus random orientation

As described in Sect. 2.2, by default, all of the theoretical calculations presented up to this point were obtained using the random orientation option of MSTM. On the one hand, we do not expect a particular orientation of the particles in the SPIN OPC to have been dominant; as a whole, the particles would have been more or less randomly oriented during the measurement. On the other hand, as an individual particle passed through the SPIN system, it would have been in some individual orientation. While we cannot assure that the fixed orientation of an individual realization that we generated would be the same as the orientation that a particular aggregate had as it passed through the SPIN system, it is still worthwhile examining how removing the random orientation option in the MSTM calculations changes the range of calculated scattered intensity values. To this end, in this section, we conduct calculations on the same realizations as in Sect. 3.5, but now with each aggregate in fixed orientation. In Table B5, we list the ranges of values of extinction, absorption, and scattering cross section for all of the realizations of the aggregate generating algorithms with each aggregate in fixed orientation. From Table B5, we see once again that the most compact aggregates (SC, IAS, PCA with $D_F$ = 2.56 and $k_{\mathrm{Sorensen}}$ = 1.186, and CCA with $D_F$ = 2.16 and $k_{\mathrm{Sorensen}}$ = 2.065) tend to have the highest extinction cross sections and scattering cross sections, which again agrees with the results of Liu and Mishchenko (2005), Liu et al. (2008), and Romshoo et al. (2021). We also see that the ranges of values in Table B5 are a little broader than the ranges of values in Table B4, as expected. The ranges of relative differences between the values used to construct Table B5 and the values used to construct Table B4 are given in Table C1. From Table C1, the relative difference in extinction cross section between fixed and random orientation reaches as high as 0.094, and this is for one of the more extended fractals with $D_F$ = 1.92, $k_{\mathrm{Sorensen}}$ = 1.873. The relative difference in absorption cross section between fixed and random orientation reaches as high as 0.066, and this is for one of the more extended fractals with $D_F$ = 2.16, $k_{\mathrm{Sorensen}}$ = 2.065. The relative difference in scattering cross section between fixed and random orientation reaches as high as 0.24, and this is for one of the more extended fractals with $D_F$ = 1.92, $k_{\mathrm{Sorensen}}$ = 1.873 for which the cross section increases from $5.803 \times 10^{-14}$ m$^2$ with random orientation to $7.189 \times 10^{-14}$ m$^2$ with fixed orientation.

In Table 6, we list the ranges of values of scattered intensity in the angular range $\theta_{\mathrm{sca}}$ = 135±20° with each aggregate in fixed orientation. As expected, overall, with fixed orientation, the ranges of the values of scattered intensity in the angular range $\theta_{\mathrm{sca}}$ = 135±20° are much broader, with the lowest value of each range significantly lower and the highest value of each range significantly higher than the respective values in Table 5, but the tendencies are the same as those seen in Table 5. As in Table 5, we see from Table 6 that the overall range of perpendicularly polarized scattered intensity in the angular range $\theta_{\mathrm{sca}}$ = 135±20° for the more extended fractal aggregates ($5.601 \times 10^{-5} - 1.965 \times 10^{-3}$) is higher than the overall range of perpendicularly polarized scattered intensity in the angular range $\theta_{\mathrm{sca}}$ = 135±20° for the more compact fractal aggregates ($5.858 \times 10^{-6} - 8.439 \times 10^{-4}$), and both of these ranges of intensity values encompass values that are more than an order of

magnitude higher than the values of perpendicularly polarized scattered intensity for the SC aggregate in fixed orientation ($2.030\times10^{-6}$) and for the IAS aggregates in fixed orientation (($1.065 - 7.332)\times10^{-6}$).

In addition, from Table 6, we can see that the highest ratio of perpendicularly to parallel polarized scattered intensity in the angular range $\theta_{\text{sca}}$ = 135±20° among all of the more compact fractal aggregates is $1.974\times10^{-1}$, and the highest ratio of perpendicularly to parallel polarized scattered intensity in the angular range $\theta_{\text{sca}}$ = 135±20° among all of the more extended fractal aggregates is $5.103\times10^{-1}$. These values are comparable to the 95th percentile values of the ratio S/P of the sample sets from the SPIN measurements (again, 0.102 and 0.515, respectively; refer to Table 1). Thus, we find that individual aggregates in fixed orientation can reproduce the highest ratios of perpendicularly to parallel polarized scattered intensity exhibited by the samples from the SPIN measurements. The ranges of relative differences between the values used to construct Table 6 and the values used to construct Table 5 are given in Table C2. From Table C2, the relative difference in $I_{\text{sca}_{\|\to\|}}\left(135\pm20°\right)$ between fixed and random orientation reaches as high as 2.1, and this is for one of the more extended fractals with $D_{\text{f}}$ = 1.92, $k_{\text{Sorensen}}$ = 1.873. The relative difference in $I_{\text{sca}_{\|\to\perp}}\left(135\pm20°\right)$ between fixed and random orientation reaches as high as 2.6, and this is for one of the more extended fractals with $D_{\text{f}}$ = 2.16, $k_{\text{Sorensen}}$ = 2.065. Notably, the relative difference in ratio of perpendicularly to parallel polarized scattered intensity between fixed and random orientation reaches as high as 17, and this is also for one of the more extended fractals with $D_{\text{f}}$ = 2.16, $k_{\text{Sorensen}}$ = 2.065 for which $\dfrac{I_{\text{sca}_{\|\to\perp}}\left(135\pm20°\right)}{I_{\text{sca}_{\|\to\|}}\left(135\pm20°\right)}$ increases from $2.766\times10^{-2}$ with random orientation to $5.103\times10^{-1}$ with fixed orientation. The relative difference in $I_{\text{sca tot}}\left(135\pm20°\right)$ between fixed and random orientation reaches as high as 2.0, and this is for one of the more extended fractals with $D_{\text{f}}$ = 1.92, $k_{\text{Sorensen}}$ = 1.873.

| Aggregate description | $I_{\text{sca}_{\|\to\|}}\left(135\pm20°\right)$ [W m$^{-2}$] | $I_{\text{sca}_{\|\to\perp}}\left(135\pm20°\right)$ [W m$^{-2}$] | Ratio of perpendicular to parallel $\dfrac{I_{\text{sca}_{\|\to\perp}}\left(135\pm20°\right)}{I_{\text{sca}_{\|\to\|}}\left(135\pm20°\right)}$ | $I_{\text{sca tot}}\left(135\pm20°\right)$ [W m$^{-2}$] |
|---|---|---|---|---|
| SC | $3.143\times10^{-2}$ | $2.030\times10^{-6}$ | $6.461\times10^{-5}$ | $3.143\times10^{-2}$ |
| IAS | $(4.400 - 4.730)\times10^{-2}$ | $(1.065 - 7.332)\times10^{-6}$ | $2.250\times10^{-5} - 1.666\times10^{-4}$ | $(4.401 - 4.730)\times10^{-2}$ |
| More compact fractal | | | | |
| CCA, $D_{\text{f}}$ = 2.34, $k_{\text{Sorensen}}$ = 1.085 | $4.232\times10^{-3} - 5.003\times10^{-2}$ | $3.420\times10^{-5} - 8.341\times10^{-4}$ | $1.739\times10^{-3} - 9.018\times10^{-2}$ | $4.337\times10^{-3} - 5.086\times10^{-2}$ |

| | | | | |
|---|---|---|---|---|
| CCA, $D_f = 2.12$, $k_{Sorensen} = 0.992$ | $(1.209 - 3.089) \times 10^{-2}$ | $6.258 \times 10^{-5} -$ $4.309 \times 10^{-4}$ | $4.579 \times 10^{-3} -$ $3.447 \times 10^{-2}$ | $(1.224 - 3.109) \times 10^{-2}$ |
| PCA, $D_f = 2.56$, $k_{Sorensen} = 1.186$ | $1.567 \times 10^{-3} -$ $4.613 \times 10^{-2}$ | $5.858 \times 10^{-6} -$ $8.439 \times 10^{-4}$ | $2.786 \times 10^{-4} -$ $1.974 \times 10^{-1}$ | $1.876 \times 10^{-3} -$ $4.690 \times 10^{-2}$ |
| More compact fractal aggregates, overall | $1.567 \times 10^{-3} -$ $5.003 \times 10^{-2}$ | $5.858 \times 10^{-6} -$ $8.439 \times 10^{-4}$ | $2.786 \times 10^{-4} -$ $1.974 \times 10^{-1}$ | $1.876 \times 10^{-3} -$ $5.086 \times 10^{-2}$ |
| More extended fractal | | | | |
| CCA, $D_f = 1.92$, $k_{Sorensen} = 1.873$ | $6.188 \times 10^{-3} -$ $4.526 \times 10^{-2}$ | $5.601 \times 10^{-5} -$ $5.430 \times 10^{-4}$ | $2.122 \times 10^{-3} -$ $4.991 \times 10^{-2}$ | $6.419 \times 10^{-3} -$ $4.565 \times 10^{-2}$ |
| CCA, $D_f = 1.68$, $k_{Sorensen} = 1.700$ | $2.212 \times 10^{-3} -$ $2.380 \times 10^{-2}$ | $6.399 \times 10^{-5} -$ $7.694 \times 10^{-4}$ | $1.066 \times 10^{-2} -$ $2.406 \times 10^{-1}$ | $2.551 \times 10^{-3} -$ $2.405 \times 10^{-2}$ |
| CCA, $D_f = 2.16$, $k_{Sorensen} = 2.065$ | $2.161 \times 10^{-3} -$ $4.644 \times 10^{-2}$ | $2.830 \times 10^{-4} -$ $1.965 \times 10^{-3}$ | $6.094 \times 10^{-3} -$ $5.103 \times 10^{-1}$ | $3.264 \times 10^{-3} -$ $4.672 \times 10^{-2}$ |
| More extended fractal aggregates, overall | $2.161 \times 10^{-3} -$ $4.644 \times 10^{-2}$ | $5.601 \times 10^{-5} -$ $1.965 \times 10^{-3}$ | $2.122 \times 10^{-3} -$ $5.103 \times 10^{-1}$ | $2.551 \times 10^{-3} -$ $4.672 \times 10^{-2}$ |

**Table 6.** Ranges of values of scattered intensity over the range $\theta_{sca} = 135 \pm 20°$ for different realizations of aggregates generated starting with an SC aggregate with $D_{outer-envelope} = 400$ nm and $d_{pp} = 35$ nm when the aggregates are in fixed orientation rather than random orientation. The values for the single SC realization from Table 2 are also included here for reference.

## 4. Discussion

As mentioned in Sects. 2.1 and 3.1, a larger value of perpendicularly polarized scattered intensity for parallel polarized incident intensity indicates some asymmetry/nonsphericity in the shape of the scattering particles or possibly birefringence or chirality in the scattering particle material. This might lead one to expect that the more fractal/extended the aggregate, the larger the value of perpendicularly polarized scattered intensity obtained. However, in computing the absolute value of scattered intensity (rather than the normalized scattering phase matrix), the elements of each scattering phase matrix are weighted by the total scattering cross section of the aggregate (refer to Eqs. (4)-(5)). Thus, the higher scattering cross sections exhibited by the more compact aggregates of each set of realizations (refer to Sect. 3.5) give more weight to their calculated scattered intensity.

We find that combining these two facts, the aggregates that possess a relatively high porosity but that are not too extended in shape are those that exhibit the highest perpendicularly polarized scattered intensity. Indeed, the realizations of the fractal aggregate generated using PCA with $D_f = 2.56$ and $k_{Sorensen} = 1.186$ exhibit the highest values of perpendicularly polarized scattered intensity of all of the more compact fractal aggregates, and the realizations of aggregate generated using CCA with

$D_f$ = 2.16 and $k_{Sorensen}$ = 2.065 exhibit the highest values of perpendicularly polarized scattered intensity of all of the more extended fractal aggregates (refer to the discussion of Fig. 7 and to Tables 5 and 6).

In addition, as presented in Sects. 3.2-3.5, we find that using the random-orientation option of MSTM on our theoretical aggregates, we are able to reproduce the qualitative behavior of the SPIN measurements when we compare to the *median values* of those measurements. Namely, the overall range of perpendicularly polarized scattered intensity in the angular range

$\theta_{sca}$ = 135±20° for the more extended theoretical fractal aggregates is consistently higher than the overall range of perpendicularly polarized scattered intensity in the angular range $\theta_{sca}$ = 135±20° for the more compact theoretical fractal aggregates.

Although the measurements and theory agree qualitatively, quantitative agreement is not always observed. As described in Sects. 3.2-3.5, we found that using the random-orientation option of MSTM on our theoretical aggregates, the *highest values*

of the ratio of perpendicularly polarized scattered intensity to parallel polarized scattered intensity in the angular range $\theta_{sca}$ = 135±20° exhibited by our theoretical aggregate realizations are not as high as the highest S/P ratios exhibited by the COJ300 and R2500U 400 nm samples from the SPIN measurements.

As shown in Sect. 3.6, only with fixed orientation do some values of the ratio of perpendicularly polarized scattered intensity to parallel polarized scattered intensity in the angular range $\theta_{sca}$ = 135±20° resemble the ratios in the 95th percentile of the

measured S/P values. In fact, for individual aggregates, an even higher measured value of the S/P ratio is possible. The bottom row of Table 1 was obtained by dividing the row labeled "S" by the row labeled "P", but if we were to present the different percentiles of S/P based on the value of S/P for individual aggregates, the 95% percentile value of S/P would

actually be ~1.0. We did not obtain a value of $\dfrac{I_{sca_{\parallel \to \perp}}\left(135 \pm 20°\right)}{I_{sca_{\parallel \to \parallel}}\left(135 \pm 20°\right)}$ close to 1.0 for any of our theoretical aggregates.

A number of reasons for the lack of quantitative agreement are possible. Foremost, we note that there could be differences

between the specifications of our theoretical aggregates and the actual chemical and physical properties of the measured aggregates. As described in Sects. 1 and 2.2, in our simulations, the aggregates in each set consist of the same number of primary particles of the same primary particle size but differing primary particle configurations. Thus, we are able to make an apples-to-apples comparison, in which all of the parameters in each set of aggregates are held constant except for the configurations of the primary particles. However, from observations (e.g., Fig. 1), the primary particle size can vary within

the same aggregate, as well as from aggregate to aggregate, and the number of primary particles can vary from aggregate to aggregate even within the same sample set. As found by Bescond et al. (2013) and Liu and Mishchenko (2005) and as reviewed in Kahnert and Kanngießer (2020), the direct backscatter depolarization ratio can vary with primary particle size and with the number of primary particles. On the other hand, Paulien et al. (2019) (also reviewed in Kahnert and Kanngießer

(2020)) found that the number of primary particles does not have a significant impact on the direct backscatter depolarization

ratio. For the cases we tested, we found that when all other parameters are held constant, the ratio $\dfrac{I_{\mathrm{sca}_{\parallel \to \perp}}(135 \pm 20°)}{I_{\mathrm{sca}_{\parallel \to \parallel}}(135 \pm 20°)}$

increases with $d_{\mathrm{pp}}$ and with $N_{\mathrm{pp}}$ for the more extended fractal aggregate ($D_{\mathrm{f}}$ = 1.92; $k_{\mathrm{Sorensen}}$ = 1.873) but not for the

more compact fractal aggregate ($D_{\mathrm{f}}$ = 2.34; $k_{\mathrm{Sorensen}}$ = 1.084). (Refer to Tables 2 and 3.) Thus, we cannot say for certain

whether further variations in $d_{\mathrm{pp}}$ and $N_{\mathrm{pp}}$ beyond what we already tested would reconcile the quantitative discrepancies.

Aside from further variations in $d_{\mathrm{pp}}$ and $N_{\mathrm{pp}}$, there could be additional differences in the configuration of the primary

particles within the aggregates beyond what our various realizations of the aggregate generating algorithms covered. When inspecting the SEM images, such as our Fig. 1, the viewing angle can mask additional asymmetry in the overall structure. Perhaps there is some chirality of shape (a slight helicity or handedness of some other form) in the aggregates examined in the SPIN measurements that the theoretical aggregate generating algorithms we employed do not fully reproduce. Alternatively, as investigated in Lu and Sorensen (1994) and Bescond et al. (2013) and as reviewed in Kahnert and

Kanngießer (2020), effects such as overlapping of primary particles and "necking" can increase the linear depolarization ratio in the direct backscattering direction to as high as 0.03, but this value is still significantly lower than the highest S/P ratios we measured. (Interestingly, Lu and Sorensen (1994) suggested necking in an attempt to reconcile the fact that their calculations underestimated the depolarization of *forward* scattered radiation.) It seems less likely, but there could also be a measure of intrinsic chirality or birefringence in the BC material used to generate the SPIN measurement samples itself; such

possible intrinsic chirality or birefringence was not considered in our theoretical calculations.

We believe the orientation of the particles throughout our experimental setup, and specifically in the detection region of the optical particle counter, is random. There remains a possibility that we do not fully understand the flow in this region and that it could lead to an organized orientation.

Yet another possibility concerns the contribution of Rayleigh scattering due to the presence of air in the chamber in the SPIN

measurements. However, this is likely to be a minor effect, due to both the weak depolarizing ability of air molecules (S/P = ~0.02; see, e.g., Sassen (2000)) and the low intensity of scattered radiation from Rayleigh scattering as compared to the intensity of scattered radiation from the aggregates, which would give the Rayleigh depolarization signal only a small weight in the overall depolarization signal. Likewise, while carbonaceous particles, such as soot, can exhibit Raman scattering (see, e.g., Le et al. (2022)), the Raman scattered signal is by nature very weak and only exhibits depolarization if the new

vibrational mode to which the molecules transition is asymmetric enough. Other technical aspects of the measurements, such as deviations of the incident wave from being a 100% coherent plane wave that is 100% polarized parallel to the scattering plane would also likely have only a minor effect.

## 5. Summary and Conclusions

Carbonaceous aerosol particles are ubiquitous in the atmosphere. Their ability to impact atmospheric chemistry, human health, and climate have led to numerous studies of their morphological, chemical, cloud formation, and radiative properties (see, e.g., Bond et al. (2006), Bond and Bergstrom (2006), Liu and Mishchenko (2018), Kahnert and Kanngießer (2020), Romshoo et al. (2021), and references therein). In this study, we analyzed laboratory measurements of scattering of visible radiation at scattering angles 135±20° by analogues of bare (uncoated) atmospheric BC aggregates obtained with the SPIN instrumentation, and using the MSTM model, we conducted theoretical calculations of scattering of visible radiation by theoretical BC aggregates constructed based on the measured morphological parameters of the laboratory generated aggregates. As discussed in Sect. 4, we found that using the random-orientation option of MSTM on our theoretical aggregates, we are able to reproduce the qualitative behavior of the SPIN measurements when we compare to the *median values* of those measurements. However, using the random-orientation option of MSTM on our theoretical aggregates, the *highest values* of the ratio of perpendicularly polarized scattered intensity to parallel polarized scattered intensity in the angular range $\theta_{sca}$ = 135±20° exhibited by our theoretical aggregate realizations are not as high as the highest S/P ratios exhibited by the COJ300 and R2500U 400 nm samples from the SPIN measurements. We found that only with fixed orientation do some values of the ratio of perpendicularly polarized scattered intensity to parallel polarized scattered intensity resemble the ratios in the 95th percentile of the measured S/P values.

We note that relatively high values of backscattering linear depolarization ratio were also obtained in the field measurements of Burton et al. (2015) (original and corrigendum). Liu and Mishchenko (2018) demonstrated that they were able to reproduce such high values of backscattering linear depolarization ratio only by simulating aged soot containing large amounts of refractory materials along with black carbon, not with bare soot. Similarly, Kahnert and Kanngießer (2020) state in their review that in most cases typical for atmospheric BC, the depolarization ratio of bare BC aggregates rarely exceeds the range 0.01-0.03. In this work, we demonstrated that even bare black carbon can exhibit high values of S/P at side/backscattering directions and that we can reproduce such high values in calculations of single scattering by bare black carbon aggregates if we use fixed orientation. However, it is important to note that the angles we inspected in the backscattering hemisphere are not the exact direct backscattering direction considered in Burton et al. (2015) and Liu and Mishchenko (2018).

There are several opportunities for future work in this area. On the experimental side, other existing instruments, such as the Droplet Measurement Technology Single Particle Soot Photometer (SP2) (Schwarz et al., 2015), or a custom-designed instrument to measure a wider range of scattering angles, might be useful in future work. Likewise, a wider range of experimental BC analogue samples, more in-depth imaging, and more sophisticated size selection would provide even better experimental statistics. On the theoretical side, additional realizations further varying the three-dimensional aggregate structure, such as varying the internal size distribution of the primary particles in each aggregate, further varying the number

of primary particles, further varying the arrangement of the primary particles, and further varying the chemical properties of the black carbon could be investigated. In addition, variations in the plane of scattering could be explored.

We conclude that our results might have important implications for remote sensing of soot aerosol via lidar backscattering, as variations in the scattering cross section, in the scattering phase function in the backscatter direction, and in the extinction cross section all potentially influence the intensity and depolarization of the lidar signal. The received photon number per

675 lidar pulse is proportional to the scattering cross section times the scattering phase function in the backscatter direction and decreases exponentially over the path of the lidar beam to and from the target as a function of the extinction cross section. Furthermore, such a direct comparison of theory to laboratory experiments of light scattering by BC aggregates as we conducted here represents an additional step towards a better overall understanding of the impact aerosol particles have on our environment and our climate system.

**Appendix A.** A summary of the terms and acronyms/symbols used in the text.

| Term | Acronym or symbol |
|---|---|
| black carbon | BC |
| simple cubic | SC |
| ideal amorphous solid | IAS |
| cluster-cluster aggregation | CCA |
| particle-cluster aggregation | PCA |
| sample set of more compact fractal aggregates from Zhang et al. (2020) | COJ300 |
| sample set of more extended fractal aggregates from Zhang et al. (2020) | R2500U |
| SPectrometer for Ice Nuclei | SPIN |
| SPIN detectors | P1, P2, S1 |
| optical particle counter | OPC |
| Droplet Measurement Technology Single Particle Soot Photometer | SP2 |
| photon count for parallel polarized incident radiation averaged over the two SPIN detectors | P=(P1+P2)/2 |
| photon count for perpendicular polarized incident radiation | S = S1 |
| total photon count parallel plus perpendicularly polarized from the SPIN detectors | P+S |
| linear depolarization ratio from the measurements | S/P |
| relative humidity | RH |
| mobility diameter | $D_m$ |
| outer-envelope diameter | $D_{outer-envelope}$ |
| number of primary particles | $N_{pp}$ |
| primary particle diameter | $d_{pp}$ |

| | |
|---|---|
| primary particle radius | $a_{pp}$ |
| fractal dimension | $D_f$ |
| fractal pre-factor as defined by Sorensen (2001) | $k_{Sorensen}$ |
| fractal pre-factor as defined by Zhang et al. (2020) | $k_{Zhang\ et\ al.}$ |
| radius of gyration | $R_g$ |
| volume-mean radius | $R_{volume-mean}$ |
| length of longest dimension of the aggregate periphery | $L_{max}$ |
| complex refractive index of the aerosol material | $m$ |
| real part of the refractive index of the aerosol material | $m_{real}$ |
| imaginary part of the refractive index of the aerosol material | $m_{imag}$ |
| multiple sphere $T$-matrix model | MSTM |
| extinction efficiency as given by MSTM | $Q_{ext\ MSTM}$ |
| absorption efficiency as given by MSTM | $Q_{abs\ MSTM}$ |
| scattering efficiency as given by MSTM | $Q_{sca\ MSTM}$ |
| extinction cross section | $\sigma_{ext}$ |
| absorption cross sections | $\sigma_{abs}$ |
| scattering cross section | $\sigma_{sca}$ |
| scattering zenith angle | $\theta_{sca}$ |
| cosine of the scattering zenith angle | $\mu_{sca}$ |
| intensity of scattered radiation as a function of scattering zenith angle | $I_{sca}\left(\theta_{sca}\right)$ |
| intensity of parallel polarized scattered radiation for parallel polarized incident radiation | $I_{sca_{\parallel\rightarrow\parallel}}$ |
| intensity of perpendicularly polarized scattered radiation for parallel polarized incident radiation | $I_{sca_{\parallel\rightarrow\perp}}$ |
| total intensity of scattered radiation | $I_{sca\ tot}$ |
| linear depolarization ratio from the calculations | $\dfrac{I_{sca_{\parallel\rightarrow\perp}}\left(135\pm20°\right)}{I_{sca_{\parallel\rightarrow\parallel}}\left(135\pm20°\right)}$ |
| scattering matrix | $\mathbf{S}$ |
| wave number | $k$ |

| | | | | | | |
|---|---|---|---|---|---|---|
| wavelength | | | | | $\lambda$ | |
| electric permittivity of the background material | | | | | $\varepsilon$ | |
| speed of light in vacuum | | | | | $c$ | |
| amplitude of the electric field of the incident electromagnetic wave | | | | | $E_0$ | |

**Table A1.** Terms and acronyms/symbols used in the text.

**Appendix B.** Tables of calculated efficiencies and cross sections.

| Aggregate description | $Q_{\text{ext MSTM}}$ | $\sigma_{\text{ext}}$ [m²] | $Q_{\text{abs MSTM}}$ | $\sigma_{\text{abs}}$ [m²] | $Q_{\text{sca MSTM}}$ | $\sigma_{\text{sca}}$ [m²] |
|---|---|---|---|---|---|---|
| starting from an SC aggregate with $D_{\text{outer-envelope}} = 300$ nm, $N_{\text{pp}} = 317$, $d_{\text{pp}} = 35$ nm | | | | | | |
| SC | 2.741 | $1.226\times10^{-13}$ | 1.792 | $8.016\times10^{-14}$ | 0.949 | $4.246\times10^{-14}$ |
| IAS | 2.780 | $1.244\times10^{-13}$ | 1.764 | $7.890\times10^{-14}$ | 1.016 | $4.546\times10^{-14}$ |
| More compact fractal: CCA, $D_{\text{f}} = 2.34$, $k_{\text{Sorensen}} = 1.085$ | 2.157 | $9.649\times10^{-14}$ | 1.629 | $7.287\times10^{-14}$ | 0.528 | $2.362\times10^{-14}$ |
| More extended fractal: CCA, $D_{\text{f}} = 1.92$, $k_{\text{Sorensen}} = 1.873$ | 2.069 | $9.254\times10^{-14}$ | 1.639 | $7.331\times10^{-14}$ | 0.430 | $1.923\times10^{-14}$ |
| starting from an SC aggregate with $D_{\text{outer-envelope}} = 400$ nm, $N_{\text{pp}} = 771$, $d_{\text{pp}} = 35$ nm | | | | | | |
| SC | 3.516 | $2.844\times10^{-13}$ | 2.065 | $1.671\times10^{-13}$ | 1.451 | $1.174\times10^{-13}$ |
| IAS | 3.448 | $2.789\times10^{-13}$ | 1.978 | $1.600\times10^{-13}$ | 1.471 | $1.190\times10^{-13}$ |
| More compact fractal: CCA, $D_{\text{f}} = 2.34$, $k_{\text{Sorensen}} = 1.085$ | 2.988 | $2.417\times10^{-13}$ | 2.091 | $1.692\times10^{-13}$ | 0.897 | $7.257\times10^{-14}$ |
| more extended fractal: CCA, $D_{\text{f}} = 1.92$, $k_{\text{Sorensen}} = 1.873$ | 2.845 | $2.301\times10^{-13}$ | 2.144 | $1.734\times10^{-13}$ | 0.701 | $5.667\times10^{-14}$ |
| starting from an SC aggregate with $D_{\text{outer-envelope}} = 500$ nm, $N_{\text{pp}} = 1529$, $d_{\text{pp}} = 35$ nm | | | | | | |
| SC | 3.981 | $5.083\times10^{-13}$ | 2.193 | $2.800\times10^{-13}$ | 1.788 | $2.283\times10^{-13}$ |
| IAS | 3.805 | $4.859\times10^{-13}$ | 2.076 | $2.651\times10^{-13}$ | 1.728 | $2.207\times10^{-13}$ |
| More compact fractal: CCA, $D_{\text{f}} = 2.34$, $k_{\text{Sorensen}} = 1.085$ | 3.776 | $4.821\times10^{-13}$ | 2.515 | $3.212\times10^{-13}$ | 1.260 | $1.609\times10^{-13}$ |
| More extended fractal: CCA, | 3.612 | $4.612\times10^{-13}$ | 2.614 | $3.338\times10^{-13}$ | 0.998 | $1.274\times10^{-13}$ |

| | | | | | | |
|---|---|---|---|---|---|---|
| $D_f = 1.92$, $k_{Sorensen} = 1.873$ | | | | | | |

**Table B1.** Values of extinction, absorption, and absorption efficiency and values of extinction, absorption, and scattering cross section for aggregates generated starting with an SC aggregate with $D_{outer\text{-}envelope}$ = 300 nm and $d_{pp}$ = 35 nm, for aggregates generated starting with an SC aggregate with $D_{outer\text{-}envelope}$ = 400 nm and $d_{pp}$ = 35 nm, and for aggregates generated starting with an SC aggregate with $D_{outer\text{-}envelope}$ = 500 nm and $d_{pp}$ = 35 nm.

| Aggregate description | $Q_{ext\ MSTM}$ | $\sigma_{ext}$ [m$^2$] | $Q_{abs\ MSTM}$ | $\sigma_{abs}$ [m$^2$] | $Q_{sca\ MSTM}$ | $\sigma_{sca}$ [m$^2$] |
|---|---|---|---|---|---|---|
| starting from an SC aggregate with $D_{outer\text{-}envelope}$ = 400 nm, $N_{pp}$ = 2106, $d_{pp}$ = 25 nm | | | | | | |
| SC | 3.436 | 2.771×10$^{-13}$ | 1.939 | 1.564×10$^{-13}$ | 1.496 | 1.207×10$^{-13}$ |
| IAS | 3.434 | 2.769×10$^{-13}$ | 1.950 | 1.573×10$^{-13}$ | 1.484 | 1.197×10$^{-13}$ |
| More compact fractal: CCA, $D_f = 2.34$, $k_{Sorensen} = 1.085$ | 2.787 | 2.248×10$^{-13}$ | 2.018 | 1.627×10$^{-13}$ | 0.769 | 6.205×10$^{-14}$ |
| More extended fractal: CCA, $D_f = 1.92$, $k_{Sorensen} = 1.873$ | 2.591 | 2.089×10$^{-13}$ | 2.064 | 1.664×10$^{-13}$ | 0.527 | 4.251×10$^{-14}$ |
| starting from an SC aggregate with $D_{outer\text{-}envelope}$ = 400 nm, $N_{pp}$ = 377, $d_{pp}$ = 45 nm | | | | | | |
| SC | 3.544 | 2.942×10$^{-13}$ | 2.075 | 1.723×10$^{-13}$ | 1.468 | 1.219×10$^{-13}$ |
| IAS | 3.476 | 2.885×10$^{-13}$ | 1.991 | 1.653×10$^{-13}$ | 1.485 | 1.232×10$^{-13}$ |
| More compact fractal: CCA, $D_f = 2.34$, $k_{Sorensen} = 1.085$ | 3.122 | 2.591×10$^{-13}$ | 2.130 | 1.768×10$^{-13}$ | 0.992 | 8.229×10$^{-14}$ |
| More extended fractal: CCA, $D_f = 1.92$, $k_{Sorensen} = 1.873$ | 3.011 | 2.499×10$^{-13}$ | 2.184 | 1.813×10$^{-13}$ | 0.826 | 6.859×10$^{-14}$ |

**Table B2.** Values of extinction, absorption, and absorption efficiency and values of extinction, absorption, and scattering cross section for aggregates generated starting with an SC aggregate with $D_{outer\text{-}envelope}$ = 400 nm and $d_{pp}$ = 25 nm and 45 nm, respectively.

| Aggregate description | $Q_{ext\ MSTM}$ | $\sigma_{ext}$ [m$^2$] | $Q_{abs\ MSTM}$ | $\sigma_{abs}$ [m$^2$] | $Q_{sca\ MSTM}$ | $\sigma_{sca}$ [m$^2$] |
|---|---|---|---|---|---|---|
| starting from an SC aggregate with $D_{outer\text{-}envelope}$ = 400 nm, $N_{pp}$ = 771, $d_{pp}$ = 35 nm; $m = 1.75 + 0.63i$ | | | | | | |
| SC | 2.627 | 2.125×10$^{-13}$ | 1.659 | 1.342×10$^{-13}$ | 0.968 | 7.831×10$^{-14}$ |

| Aggregate description | $Q_{ext}$ | $\sigma_{ext}$ | $Q_{abs}$ | $\sigma_{abs}$ | $Q_{sca}$ | $\sigma_{sca}$ |
|---|---|---|---|---|---|---|
| IAS | 2.650 | $2.144\times10^{-13}$ | 1.629 | $1.318\times10^{-13}$ | 1.021 | $8.260\times10^{-14}$ |
| More compact fractal: CCA, $D_f$ = 2.34, $k_{Sorensen}$ = 1.085 | 2.148 | $1.737\times10^{-13}$ | 1.626 | $1.315\times10^{-13}$ | 0.522 | $4.221\times10^{-14}$ |
| More extended fractal: CCA, $D_f$ = 1.92, $k_{Sorensen}$ = 1.873 | 2.060 | $1.666\times10^{-13}$ | 1.660 | $1.343\times10^{-13}$ | 0.400 | $3.236\times10^{-14}$ |
| starting from an SC aggregate with $D_{outer-envelope}$ = 400 nm, $N_{pp}$ = 771, $d_{pp}$ = 35 nm; $m=1.85+0.71i$ | | | | | | |
| SC | 2.912 | $2.356\times10^{-13}$ | 1.770 | $1.432\times10^{-13}$ | 1.143 | $9.243\times10^{-14}$ |
| IAS | 2.923 | $2.365\times10^{-13}$ | 1.730 | $1.399\times10^{-13}$ | 1.193 | $9.653\times10^{-14}$ |
| More compact fractal: CCA, $D_f$ = 2.34, $k_{Sorensen}$ = 1.085 | 2.348 | $1.899\times10^{-13}$ | 1.718 | $1.390\times10^{-13}$ | 0.629 | $5.090\times10^{-14}$ |
| More extended fractal: CCA, $D_f$ = 1.92, $k_{Sorensen}$ = 1.873 | 2.237 | $1.809\times10^{-13}$ | 1.753 | $1.418\times10^{-13}$ | 0.484 | $3.914\times10^{-14}$ |
| starting from an SC aggregate with $D_{outer-envelope}$ = 400 nm, $N_{pp}$ = 771, $d_{pp}$ = 35 nm; $m=2.26+1.26i$ | | | | | | |
| SC | 4.056 | $3.281\times10^{-13}$ | 2.225 | $1.800\times10^{-13}$ | 1.831 | $1.481\times10^{-13}$ |
| IAS | 3.898 | $3.154\times10^{-13}$ | 2.104 | $1.702\times10^{-13}$ | 1.794 | $1.451\times10^{-13}$ |
| More compact fractal: CCA, $D_f$ = 2.34, $k_{Sorensen}$ = 1.085 | 3.419 | $2.766\times10^{-13}$ | 2.201 | $1.781\times10^{-13}$ | 1.218 | $9.852\times10^{-14}$ |
| More extended fractal: CCA, $D_f$ = 1.92, $k_{Sorensen}$ = 1.873 | 3.196 | $2.586\times10^{-13}$ | 2.242 | $1.814\times10^{-13}$ | 0.954 | $7.721\times10^{-14}$ |

**Table B3.** Values of extinction, absorption, and absorption efficiency and values of extinction, absorption, and scattering cross section for aggregates generated starting with an SC aggregate with $D_{outer-envelope}$ = 400 nm and $d_{pp}$ = 35 nm with varying primary particle refractive indices.

| Aggregate description | Range of $\sigma_{ext}$ values [m²] | Range of $\sigma_{abs}$ values [m²] | Range of $\sigma_{sca}$ values [m²] |
|---|---|---|---|
| SC | $2.844\times10^{-13}$ | $1.671\times10^{-13}$ | $1.174\times10^{-13}$ |
| IAS | $(2.788 - 2.790)\times10^{-13}$ | $(1.598 - 1.600)\times10^{-13}$ | $(1.190 - 1.190)\times10^{-13}$ |
| More compact fractal | | | |
| CCA, $D_f$ = 2.34, $k_{Sorensen}$ = 1.085 | $(2.404 - 2.417)\times10^{-13}$ | $(1.684 - 1.695)\times10^{-13}$ | $(7.144 - 7.290)\times10^{-14}$ |
| CCA, $D_f$ = 2.12, $k_{Sorensen}$ = 0.992 | $(2.241 - 2.258)\times10^{-13}$ | $(1.733 - 1.747)\times10^{-13}$ | $(4.975 - 5.202)\times10^{-14}$ |

| Aggregate description | Range of $\sigma_{ext}$ values | Range of $\sigma_{abs}$ values | Range of $\sigma_{sca}$ values |
|---|---|---|---|
| PCA, $D_f$ = 2.56, $k_{Sorensen}$ = 1.186 | $(2.600 - 2.635) \times 10^{-13}$ | $(1.682 - 1.689) \times 10^{-13}$ | $(9.186 - 9.470) \times 10^{-14}$ |
| More extended fractal | | | |
| CCA, $D_f$ = 1.92, $k_{Sorensen}$ = 1.873 | $(2.287 - 2.321) \times 10^{-13}$ | $(1.727 - 1.742) \times 10^{-13}$ | $(5.551 - 5.932) \times 10^{-14}$ |
| CCA, $D_f$ = 1.68, $k_{Sorensen}$ = 1.700 | $(2.152 - 2.180) \times 10^{-13}$ | $(1.784 - 1.791) \times 10^{-13}$ | $(3.618 - 3.961) \times 10^{-14}$ |
| CCA, $D_f$ = 2.16, $k_{Sorensen}$ = 2.065 | $(2.500 - 2.531) \times 10^{-13}$ | $(1.700 - 1.710) \times 10^{-13}$ | $(7.950 - 8.309) \times 10^{-14}$ |

**Table B4.** Ranges of values of extinction, absorption, and absorption cross section for different realizations of aggregates generated starting with an SC aggregate with $D_{outer-envelope}$ = 400 nm and $d_{pp}$ = 35 nm. The values for the single SC realization from Table B1 are also included here for reference.

| Aggregate description | Range of $\sigma_{ext}$ values [m²] | Range of $\sigma_{abs}$ values [m²] | Range of $\sigma_{sca}$ values [m²] |
|---|---|---|---|
| SC | $2.849 \times 10^{-13}$ | $1.676 \times 10^{-13}$ | $1.173 \times 10^{-13}$ |
| IAS | $(2.788 - 2.792) \times 10^{-13}$ | $(1.599 - 1.601) \times 10^{-13}$ | $(1.189 - 1.191) \times 10^{-13}$ |
| More compact fractal | | | |
| CCA, $D_f$ = 2.34, $k_{Sorensen}$ = 1.085 | $(2.326 - 2.565) \times 10^{-13}$ | $(1.651 - 1.746) \times 10^{-13}$ | $(6.663 - 8.181) \times 10^{-14}$ |
| CCA, $D_f$ = 2.12, $k_{Sorensen}$ = 0.992 | $(2.145 - 2.319) \times 10^{-13}$ | $(1.717 - 1.769) \times 10^{-13}$ | $(4.250 - 5.498) \times 10^{-14}$ |
| PCA, $D_f$ = 2.56, $k_{Sorensen}$ = 1.186 | $(2.557 - 2.765) \times 10^{-13}$ | $(1.646 - 1.742) \times 10^{-13}$ | $9.102 \times 10^{-14} - 1.024 \times 10^{-13}$ |
| More extended fractal | | | |
| CCA, $D_f$ = 1.92, $k_{Sorensen}$ = 1.873 | $(2.213 - 2.524) \times 10^{-13}$ | $(1.695 - 1.805) \times 10^{-13}$ | $(4.735 - 7.189) \times 10^{-14}$ |
| CCA, $D_f$ = 1.68, $k_{Sorensen}$ = 1.700 | $(2.131 - 2.190) \times 10^{-13}$ | $(1.744 - 1.821) \times 10^{-13}$ | $(3.347 - 4.463) \times 10^{-14}$ |
| CCA, $D_f$ = 2.16, $k_{Sorensen}$ = 2.065 | $(2.359 - 2.734) \times 10^{-13}$ | $(1.645 - 1.813) \times 10^{-13}$ | $(7.122 - 9.213) \times 10^{-14}$ |

**Table B5.** Ranges of values of extinction, absorption, and absorption cross section for different realizations of aggregates generated starting with an SC aggregate with $D_{outer-envelope}$ = 400 nm and $d_{pp}$ = 35 nm when the aggregates are in fixed orientation rather than random orientation. The values for the single SC realization in fixed orientation are also included here for reference.

**Appendix C.** Relative differences between fixed and random orientation.

| Aggregate description | Range of relative difference values for $\sigma_{\text{ext}}$ | Range of relative difference values for $\sigma_{\text{abs}}$ | Range of relative difference values for $\sigma_{\text{sca}}$ |
|---|---|---|---|
| SC | 0.0017 | 0.0035 | 0.00090 |
| IAS | 0.00020 – 0.0015 | 0.00010 – 0.0019 | 0.00088 – 0.0014 |
| More compact fractal | | | |
| CCA, $D_{\text{f}}$ = 2.34, $k_{\text{Sorensen}}$ = 1.085 | 0.0075 – 0.061 | 0.0044 – 0.032 | 0.013 – 0.13 |
| CCA, $D_{\text{f}}$ = 2.12, $k_{\text{Sorensen}}$ = 0.992 | 0.0011 – 0.045 | 0.0011 – 0.014 | 0.018 – 0.17 |
| PCA, $D_{\text{f}}$ = 2.56, $k_{\text{Sorensen}}$ = 1.186 | 0.0020 – 0.049 | 0.0023 – 0.033 | 0.0017 – 0.082 |
| More extended fractal | | | |
| CCA, $D_{\text{f}}$ = 1.92, $k_{\text{Sorensen}}$ = 1.873 | 0.012 – 0.094 | 0.0031 – 0.045 | 0.057 – 0.24 |
| CCA, $D_{\text{f}}$ = 1.68, $k_{\text{Sorensen}}$ = 1.700 | 0.00030 – 0.0094 | 0.0027 – 0.023 | 0.0015 – 0.13 |
| CCA, $D_{\text{f}}$ = 2.16, $k_{\text{Sorensen}}$ = 2.065 | 0.0060 – 0.080 | 0.0028 – 0.066 | 0.040 – 0.12 |

**Table C1.** Range of values of relative difference in the values of extinction, absorption, and absorption cross section between fixed orientation and random orientation for the different realizations of aggregates generated starting with an SC aggregate with $D_{\text{outer-envelope}}$ = 400 nm and $d_{\text{pp}}$ = 35 nm. Each value of relative difference is calculated as the absolute value of (fixed – random)/random and is listed to two significant figures.

| Aggregate description | Range of relative difference values for $I_{\text{sca}_{\parallel \to \parallel}}\,(135 \pm 20°)$ | Range of relative difference values for $I_{\text{sca}_{\parallel \to \perp}}\,(135 \pm 20°)$ | Range of relative difference values for $\dfrac{I_{\text{sca}_{\parallel \to \perp}}\,(135 \pm 20°)}{I_{\text{sca}_{\parallel \to \parallel}}\,(135 \pm 20°)}$ | Range of relative difference values for $I_{\text{sca tot}}\,(135 \pm 20°)$ |
|---|---|---|---|---|
| SC | 0.082 | 0.86 | 0.84 | 0.083 |
| IAS | 0.016 – 0.070 | 0.47 – 0.89 | 0.46 – 0.90 | 0.016 – 0.070 |
| More compact fractal | | | | |
| CCA, $D_{\text{f}}$ = 2.34, $k_{\text{Sorensen}}$ = 1.085 | 0.032 – 1.9 | 0.38 – 1.1 | 0.12 – 2.6 | 0.016 – 1.8 |
| CCA, $D_{\text{f}}$ = 2.12, $k_{\text{Sorensen}}$ = 0.992 | 0.14 – 0.92 | 0.21 – 0.78 | 0.15 – 1.1 | 0.13 – 0.90 |

| | | | | |
|---|---|---|---|---|
| PCA, $D_{\mathrm{f}}$ = 2.56, $k_{\mathrm{Sorensen}}$ = 1.186 | 0.20 – 1.1 | 0.12 – 0.98 | 0.34 – 6.0 | 0.19 – 1.1 |
| More compact fractal aggregates, overall | 0.032 – 1.9 | 0.21 – 1.1 | 0.12 – 6.0 | 0.016 – 1.8 |
| More extended fractal | | | | |
| CCA, $D_{\mathrm{f}}$ = 1.92, $k_{\mathrm{Sorensen}}$ = 1.873 | 0.090 – 2.1 | 0.17 – 0.87 | 0.62 – 1.7 | 0.073 – 2.0 |
| CCA, $D_{\mathrm{f}}$ = 1.68, $k_{\mathrm{Sorensen}}$ = 1.700 | 0.030 – 0.89 | 0.14 – 1.4 | 0.25 – 13 | 0.034 – 0.87 |
| CCA, $D_{\mathrm{f}}$ = 2.16, $k_{\mathrm{Sorensen}}$ = 2.065 | 0.32 – 1.1 | 0.43 – 2.6 | 0.15 – 17 | 0.29 – 1.1 |
| More extended fractal aggregates, overall | 0.030 – 2.1 | 0.14 – 2.6 | 0.15 – 17 | 0.034 – 2.0 |


**Table C2.** Range of values of relative difference in the values of scattered intensity over the range $\theta_{\mathrm{sca}}$ = 135±20° between fixed orientation and random orientation for the different realizations of aggregates generated starting with an SC aggregate with $D_{\mathrm{outer\text{-}envelope}}$ = 400 nm and $d_{\mathrm{pp}}$ = 35 nm. Each value of relative difference is calculated as the absolute value of (fixed – random)/random and is listed to two significant figures.


**Funding:** This work was supported by the MIT International Science and Technology Initiatives (MISTI)-Israel Seed Fund (2017-2018); Carynelisa Haspel is supported by the Israel Science Foundation (1872/17 and 2187/21).

**Code/Data availability:** The data and model output from this study are available from the authors by request.

**Author contribution:** CZ and MJW carried out the laboratory experiments under the supervision of DJC. CH performed the calculations and prepared the manuscript with contributions from all co-authors. DJC and CH acquired the funding for the project leading to this publication.


**Competing interests:** The authors declare that they have no conflict of interest.

**Acknowledgements:** We thank Daniel Mackowski for supplying the aggregate generating algorithm and for consultation on its use, as well as for helpful discussions on MSTM in general; Ynon Hefets for helpful discussions on polarization and

suggestions for future simulations; Ofir Shoshanim and Jonas Gienger for helpful discussions on scattering experiments; Droplet Measurement Technologies and Sarvesh Garimella for useful information on the SPIN instrument; and Yue Zhang

and Leonid Nichman for assistance with the black carbon measurements. We also thank the two anonymous referees for their helpful suggestions.

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
