# Peer review of "Measurements and Calculations of Enhanced Side/Back Scattering of Visible Radiation by Black Carbon Aggregates"

_EGUsphere, 2023_

## Author Comment (AC1)

**Reviewer 1**

General comments:

The authors of this study examined the depolarization ratio of pure black carbon fractal aggregates. Based on previous experimental data, the depolarization ratio (DPR) has been modeled and the model has been compared to the measured DPR obtained through SPIN measurements. It has been demonstrated that using random orientation in T-matrix models results in an underestimation of the DPR. The purpose of this article is to highlight the importance of using fixed orientation when modeling the optical properties of BC fractal aggregates. The findings of this study are original in that they extend the previous work done by Sela and Haspel (2021), and there is a novelty to the analysis. I would like to provide some comments on the paper, and once these comments have been addressed, I recommend it for publication.

**Response:** We thank Reviewer 1 and believe his/her comments have helped us substantially improve the manuscript. Please find our changes and point by point responses below.

Specific comments:

1.  The authors begin their introduction by describing previous research. In my opinion, however, it is missing context about why we need to investigate the DPR of soot particles and how they might be relevant to remote sensing or radiative forcing estimates.

    **Response:** Based on this comment and comment 3 of Reviewer 2, we now begin section 1 with a new paragraph introducing the importance of the single scattering properties of BC/soot particles and their linear depolarization ratio.

2.  It would be better to move some information, in particular the mathematical equations in Section 2.2, to the Appendix about theoretical methods in order to make the publication easier to read.

    **Response:** We think that the mathematical expressions are a fundamental part of the methods, and therefore, similar to the equations in the methods section of the Romshoo et al. (2021) paper that the reviewer mentions in his/her comment 7, we prefer to leave them in the methods section. However, as we mention in response to comment 6 below, we have moved five of the tables to Appendix B as per both reviewers' suggestions, which should make the manuscript easier to read.

3.  The difference between eq. 8 and eq. 9 is not clear. The optical efficiency can be calculated as the ratio between the optical cross-section and the geometric cross-section. You have taken the ratio of two geometric cross-sections in equation 8.

    **Response:** We agree that the distinction is unclear, and we have decided that what was labeled Equation 8 is unnecessary. We have now removed what was labeled Equation 8, and we have adjusted the wording accordingly.

4.  In the results of the total scattered intensity over theta-sca (for example, Figs. 4, 6, 7 You may also so on), the authors have pointed out the differences in soot characteristics using terms such as SC_300_35, etc. There is an overwhelming amount of terminology used in the caption of these results, which makes it difficult for the reader to understand them. In each figure, it is recommended to indicate the point of difference within the panels. For examples in Fig. 4, the difference between the panels is the change in the mobility diameter. A description of the corresponding mobility diameter can be included in each panel by the authors. And specify the other common properties as a single line in the caption. Similarly, in the discussion, the authors can specify the relevant parameter that they are discussing in each section (rather than using long technical terms). The second part of this comment is a choice of style. The authors can choose to remain with their style.

    **Response:** In response to comment 10 of Reviewer 2 and in response to this comment, we have now indicated the point of difference within the panels of Figures 4-7, and we have now reworded the captions. In fact, we have removed all of the aggregate labels from the manuscript and replaced them with descriptions of the aggregates. What was Table 1 is no longer necessary and has been removed. (We have also changed the color scheme of the curves to the CUD color scheme in https://jfly.uni-koeln.de/color/#convert.)

5.  The authors have discussed the difference between the modelled DPR and the measured DPR at the end of section 3.2. This was done by comparing the DPR from Table 2 with those from Table 4. It is an important result. Ideally, the differences should be visualized as a 1:1 corelation with a linear regression fit.

    **Response:** We believe that this is an interesting suggestion, and we considered how to address it. From what is now Table 1, using the 50% percentile values, there are unfortunately only two values that could have been used in the suggested correlation. From what is now Table 2, with our default aggregates, there are only four values that could have been used in the suggested correlation. The lack of data for building a correlation ultimately led us to conclude that a correlation/regression fit would not provide the desired results. Pending future studies, we do believe this is a good suggestion, but ultimately we concluded that it could not be implemented here. Given the lack of data, we feel the direct comparison in the text remains the best presentation of results.

6.  The results, particularly the tables about optical properties, should be moved to the Appendix or Supplementary sections.

    **Response:** Reviewer 2 made a similar suggestion about moving some results to an appendix. We have now moved all of the tables of efficiencies and cross sections from all of the sections to Appendix B.

7.  In Table 9, the authors provide the range of optical properties for different realisations of pure BC aggregates. It would be preferable if the range was expressed as a percentage change. The study by Romshoo et al., 2021 (Fig. 4) has also demonstrated that there is variability in the optical properties of pure BCFAs at 660 nm for 30 realisations at different

mobility diameters and fractal dimensions. What is the comparison between your results and theirs? Romshoo, B., Müller, T., Pfeifer, S., Saturno, J., Nowak, A., Ciupek, K., Quincey, P., and Wiedensohler, A.: Optical properties of coated black carbon aggregates: numerical simulations, radiative forcing estimates, and size-resolved parameterization scheme, Atmos. Chem. Phys., 21, 12989–13010, https://doi.org/10.5194/acp-21-12989-2021, 2021.

**Response:** We thank the reviewer for calling our attention to the Romshoo et al. (2021) study. We now cite Romshoo et al. (2021) in five places in the manuscript. In particular, we have added a citation of Romshoo et al. (2021) in the places in which we state that the extinction and scattering cross sections of soot aggregates increase as the aggregates become more compact, which can be seen clearly from their Figure 4. This includes in our discussion of our Table B4 (previously our Table 9). However, we do not think that it would be correct to attempt a direct quantitative comparison of our Table B4 with Figure 4 of Romshoo et al., as the parameters used are not the same. (Some parameters are similar, but actually none are identical. The wavelength in Figure 4 of Romshoo et al. is 660 nm, whereas we used 670 nm. The refractive index used in Figure 4 of Romshoo et al. is $2.0 + 0.63i$, whereas we used $2.0 + 1.0i$, with higher absorption. The mobility diameters in Figure 4 of Romshoo et al. are 150 nm, 250 nm, 500 nm, and 1000 nm, whereas our default diameter is 400 nm; the closest comparison would be to mobility diameter 500 nm. The primary particle diameter in Romshoo et al. is 30 nm, whereas our default primary particle diameter is 35 nm. The number of primary particles in Romshoo et al. is set via their Equation 4, whereas the number of primary particles in our default case is 771; our number of primary particles is higher. The fractal dimensions in Figure 4 of Romshoo et al. vary from 1.5 to 2.2 in steps of 0.1. Our values are 1.68, 1.92, 2.12, 2.16, 2.34, 2.56; the closest comparison would be to fractal dimensions 1.7, 1.9, and 2.1, while 2.34 and 2.56 are beyond the range explored in Romshoo et al.).

Regarding expressing the values shown in Table B4 (previously Table 9) as a percentage change, it is not clear to us with respect to what the percent difference should be referenced. If the reviewer could clarify which value, we can consider this change. Note also our response to comment 8 below; perhaps that already covers what the reviewer intended.

8. Similarly to comment 7, it would be more beneficial to have the range of optical properties and the DPR expressed as a percent change for random vs fixed orientation. It is easier for others to use or cite this information.

   **Response:** We now include two tables of relative differences (fixed orientation minus random orientation divided by random orientation) in Appendix C, one for the cross sections and one for the intensities.

9. The authors mentioned the atmospheric implications in the conclusion, but no discussion/result has been provided. Would it be possible for the authors to explain how the DPR range (random vs fixed) affects the calculation of lidar backscattering? The sensitivity study can be demonstrated using a simple analytical equation, is there one?

**Response:** We agree that more of an explanation was warranted here. The first sentence of the last paragraph of the Summary and Conclusions now reads, "We conclude that our results might have important implications for remote sensing of soot aerosol via lidar backscattering, as variations in the scattering cross section, in the scattering phase function in the backscatter direction, and in the extinction cross section all potentially influence the intensity and depolarization of the lidar signal. The received photon number per lidar pulse is proportional to the scattering cross section times the scattering phase function in the backscatter direction and decreases exponentially over the path of the lidar beam to and from the target as a function of the extinction cross section."

A quantitative demonstration of the sensitivity, however, would involve many assumptions regarding variables that we did not use or present in this study. (See the paragraph below.) Therefore, we have not included a quantitative demonstration.

The received photon number per lidar pulse from range $z$ at wavelength $\lambda$, $N_{\mathrm{rec}_\lambda}(z)$, is proportional to: $\left(\beta_{\mathrm{mol}_\lambda} + \beta_{\mathrm{aer}_\lambda}\right)\exp\left(-2\int_0^z \left(\alpha_{\mathrm{mol}_\lambda}(z') + \alpha_{\mathrm{aer}_\lambda}(z')\right)dz'\right)$,

where $\alpha$ is the extinction coefficient ($\alpha_{\mathrm{aer}_\lambda} = n_{\mathrm{aer}}\sigma_{\mathrm{ext\,aer}_\lambda}$), $n_{\mathrm{aer}}$ is the number concentration of aerosol particles, $\sigma_{\mathrm{ext\,aer}_\lambda}$ is the extinction cross section of the aerosol particles, and $\beta$ is the backscatter coefficient ($\beta_{\mathrm{aer}_\lambda} = n_{\mathrm{aer}}\sigma_{\mathrm{sca\,aer}_\lambda}P_{\mathrm{aer}_\lambda}(180°)$, where $\sigma_{\mathrm{sca\,aer}_\lambda}$ is the scattering cross section of the aerosol particles, and $P_{\mathrm{aer}_\lambda}(180°)$ is the value of the scattering phase function in the direct backscatter direction; there is possibly an additional division by $4\pi$ depending on the normalization of the phase function). Even neglecting $\alpha_{\mathrm{mol}_\lambda}$ and $\beta_{\mathrm{mol}_\lambda}$, the calculation requires assuming a vertical profile of the aerosol particles. Even if we assume that the backscatter from the particles is from a single homogeneous aerosol layer, we would still need to assume a value for $n_{\mathrm{aer}}$ and a value for the geometric depth of the aerosol layer. We would also be implicitly presuming an aerosol layer comprised of only BC particles of the sizes that we inspected in this study, or else we would need to make some assumptions regarding other sizes and compositions of particles.

10. The SPIN measures at 135 degrees, which is why the authors were able to compare the modelling and measurements of DPR for a single wavelength. Would it be possible for the authors to mention in the discussion whether there are other instruments that can be used to measure other angles or all angles? Is it possible to improve the understanding of this subject by using such instruments?

    **Response:** The reviewer brings up a good point. The SPIN detector was used in this work, because we had a complementary study on BC ice nucleation. There are other instruments, or an instrument could be custom-built, to make measurements over a different or greater angular range. We have now added the following sentence to the conclusions: "On the experimental side, other existing instruments, such as the Droplet Measurement Technology Single Particle Soot Photometer (SP2) (Schwarz et al., 2015), or a customdesigned instrument to measure a wider range of scattering angles, might be useful in future work."

Technical corrections:

1.  Line 53 – "vein" do the authors mean "way"

    **Response:** We think that "in a similar vein" is a commonly used expression, but it is not necessary. We have changed "vein" to "manner" (in two places).

2.  Line 92 – Section 2.1 instead of "section 1"

    **Response:** Actually, the mean microphysical properties of the COJ300 and R2500U 400 nm samples from Zhang et al. (2020) are listed in section 1.

3.  Line 41-44 – sometimes it is better to write two sentences instead of one long sentence

    **Response:** The sentence has been broken into two sentences, as suggested.

4.  Captions of figures – the entire text does not need to be in Bold

    **Response:** The captions of the figures have been switched from bold to regular text.

5.  At the end of the manuscript, in the Appendix, please provide the list of terms/abbreviations

    **Response:** A list of terms/acronyms/symbols is now provided in Appendix A, as suggested.

**Reviewer 2**

Haspel et al. present experimental measurements and theoretical calculations of polarized scattered light for different types of BC aggregates. The measurements (2 polarization states, integrated over a scattering angle range from 115-155°) are performed for 2 types of 400 nm size-selected BC aggregates. The theoretical calculations (2 polarization states, angularly resolved plus integrated quantities) are performed using the multiple sphere T-matrix model for a large range of different BC aggregates (varying order/disorder, fractal dimension, number and diameter of primary spheres, aggregate orientation, refractive indices). The goal is not to perform a full quantitative intercomparison between the measurements and calculations, but rather to to see whether they follow the same qualitative tendencies. It is shown that they largely do. A limited quantitative comparison is also perfomed with respect to measured and calculated linear depolorization ratios.

Theoretical calculations of light scattering by BC aggregates are only rarely compared with actual measurements. Therefore, I think this manuscript will be a useful contribution to the literature. However, there are a number of important issues that I believe should be addressed before the manuscript is accepted for publication.

**Response:** We thank Reviewer 2 and believe his/her comments have helped us substantially improve the manuscript. Please find our changes and point by point responses below.

1. L23-26: There are many potential reasons for the discrepancies between measured and calculated polarization ratios (as summarized nicely in Section 4). I am not convinced that this is the most likely explanation and therefore this part of the abstract reads to me as an over-interpretation of the results.

   **Response:** Based on this comment, the sentence, "Thus, we suggest that it is possible that models of scattering by BC aggregates that employ the random orientation assumption/option may underpredict the linear depolarization ratio of actual BC aggregates" has now been removed from the abstract.

2. L38: Throughout the paper there is conflation of the terms aggregate 'order/disorder' and 'fractalness'. For example, here is it is said that the authors will examine how the degree of aggregate disorder affects actual measurements. But the measurements are performed on aggregates of different fractal dimension but unkownn order, and ultimately they are then compared with the CCA aggregate calculations. It would be helpf ful if the differences (and similarities) between these two concepts were more clearly described and if more care was taken not to conflate the 2 terms throughout the paper.

   **Response:** We agree. In response to this comment, we no longer frame the study as an examination of the effect of disorder, but rather as an examination of the effect of the variation in the configuration of the primary particles, holding all other parameters constant to the greatest possible extent. We have decided that a careful description of the distinction and/or connection between disorder and fractalness (as described, e.g., in Bunde, A., and S.

Havlin, Eds., Fractals and Disordered Systems, Springer-Verlag, Berlin, 1991) is beyond the scope of this manuscript and is unnecessary for conveying the results we present.

3. L41: The obvious atmospheric connection here is that the more fractal-like aggregates better represent fresh BC close to emission sources, while the more-spherical aggregates represent aged BC that has undergone restructuring. Perhaps this should be discussed here in order to better highlight the atmospheric relevance of the work. Can it also be said (with reference to past studies) that collapsed, spherical-like BC particles in the atmosphere are likely to be better represented by the IAS than the SC aggregates?

   **Response:** We agree with the reviewer that the compact fractal aggregates and the IAS aggregates are both better models of collapsed BC than the SC aggregates, but we do not believe that we can say this with absolute certainty. To address this, and based also on comment 1 of Reviewer 1, we have now added a new paragraph to the beginning of the introduction. This new paragraph includes the sentence, "More extended fractal aggregates are generally considered to be analogues of relatively fresh/unaged black carbon (BC), while more compact, roughly spherical aggregates are considered to be analogues for BC that has "collapsed" into a quasi-spherical structure after cloud processing or aging (Ma et al., 2013; Sedlaceck et al., 2015)."

4. Fig. 1: Are these SEM images of aggregates that have been size-selected? There appears to be quite some variability in the aggregate sizes. Also panel (a) shows aggregates that have coagulated into pairs. This will strongly affect their scattering properties.

   **Response:** We thank the reviewer for bringing this to our attention. Yes, the SEM images show 400 nm aggregates size-selected using a differential mobility analyzer (DMA; model 2002; Brechtel Manufacturing Inc) with a 500 nm impactor at the DMA inlet. The size-selected aggregates were then collected using a micro-orifice uniform deposit impactor (MOUDI; model M135-10; TSI Incorporated). The variability of aggregate sizes could be attributed to the difference between the mobility (DMA) and aerodynamic (MOUDI) diameters. The grey smaller aggregates in the upper part of panel (a) might be fractures of soot agglomerates/aggregates due to impaction. When we collect the samples with MOUDI, there is a chance that aggregates with similar inertia could pile on and stick to one another. This will lead to coagulated pairs in the 2D SEM image in panel (a). However, the aggregates are more likely to be separated in the aerosol stream.

   To clarify that the aggregates in the images have been size selected, we have now changed the caption of Figure 1 to "SEM images of size-selected 400-nm aggregates from sample set (a) COJ300 and (b) R2500U."

5. L74: A diagram would help grealy to visualize the positions of the different detectors.

   **Response:** We agree. However, rather than add a diagram to our manuscript, we now refer the reader to Garimella et al. (2016), in which the SPIN OPC is described in greater detail, along with a diagram. We have added, "See Garimella et al. (2016) for a more complete description of the OPC geometry, including a diagram of the instrumentation."

6. L88: These thesholds are high in terms of diameter (I assume optical diameter as measured by the side-scattering detector in the SPIN). It seems likely that some doubly-charged particles would still be included in the analysis? Is there a reason these thresholds were set so high?

**Response:** Yes, the thresholds here are chosen based on the optical diameter measured by the side-scattering signal. Since the particles are size-selected with the DMA, we have to take the shape factor and effective density of the aerosols into account when comparing the optical and mobility sizes. For extended fractal aggregates like R2500U, the effective density would be much lower than for spherical aggregates with no voids, and the shape factor could be larger than 1, leading to larger optical diameters. As stated here, we set the threshold following a consistent 90% percentile criterion in order to keep as much data as possible. There could be doubly-charged particles in the analysis, for which we offer not only mean/median values, but also different percentiles from 5 to 95 in what is now Table 1 to give the reader an idea of how broad these scattering signals span.

In response to this comment, we have amended the text to indicate that size thresholds were applied to the optical diameter data and that the thresholds are chosen to account for the differences between the optical and mobility diameter while mitigating the impact of multiply charged particles:

"To avoid the influence of multiply charged BC particles, which could reach up to 16% of the total BC population, size thresholds corresponding to the 90% quantile of optical diameter (1310.7 nm for COJ300 and 6769.4 nm for R2500J, respectively) were applied to the particle-by-particle data. This filter accounts for the differences between the optical and mobility diameter while minimizing the impact of doubly and triply charged particles in our data analysis."

7. Table 1: The aggregate labels are difficult to take in and readibility later on becomes very laborious (e.g. comparing CCA_2.34_1.085_400_35 with CCA_1.92_1.085_400_35). I encourage the authors to think carefully about a better aggregate labelling system to improve the readibility of the text.

**Response:** In response to comment 4 of Reviewer 1 and in response to this comment, we have removed all of the aggregate labels from the manuscript and replaced them with descriptions of the aggregates. What was Table 1 is no longer necessary and has been removed.

8. Table 2: Perhaps my biggest overall criticism of the manuscript is that it is difficult to get a sense of the measurement uncertainties and how much can be interpreted from the differences observed. For example, there is clear systematic bias (~20%) between detectors P1 and P2. Why could this be? Could a similar bias be affecting detector S1 in relation to P1 or P2, and therefore the S/P ratios? Would it perhaps be better to use only P1 in the S/P ratio, since P1 and S1 are at least measuring the same angular position? In addition, measurements are only shown for BC aggregates of complex morphology. Did the authors

also measure some simpler test aerosols that were either spherical or closer-to-spherical? It would be interesting to see the S/P statistics for such aerosols to get a sense of how significant the differences are between the COJ300 and R2500U sets.

**Response:** We thank the reviewer for highlighting the importance of considering uncertainty in our calculations. Note that in what is now Table 1, we present values at the 5th, 25th, 50th, 75th, and 95th percentiles in order to give readers a sense of the uncertainty in the data values. We have clarified this point in the revised text:

"A summary of the scattering measurements from the SPIN OPC is given in Table 1. The measurements are presented for the 5th, 25th, 50th, 75th, and 95th percentiles in order to demonstrate the scope of the variability and uncertainty in the data."

The reviewer also mentions a difference in the data between the P1 and P2 sensors. As can be seen from the values at different percentiles, any systematic bias between the P1 and P2 sensors is lower than the variability in the photon counts. We consider any potential bias of this sort to be a negligible source of uncertainty in our calculations. We now state this in the text:

"We note that the P1 photon counts are higher than the P2 photon counts. However, this possibly systematic difference is small compared with the variability in photon counts from particle to particle."

9.   Table 3: The paper contains a lot of data tables that take substantial effort to comprehend. The presentation quality could be improved a lot, sometimes just with simple formatting. For example, in this table it would be useful to visually separate the different D_outer-envelope sets, even just with simple horizontal lines.

**Response:** We have now moved all of the tables of efficiencies and cross sections from all of the sections to Appendix B. In addition, we have now added horizontal lines to all of the tables to separate groups of simulations from one another, as suggested.

10.  Figure 4 (and other similar figures): the figure captions are overly difficult to comprehend. Revision is required. A key piece of information is missing: what are the thick solid lines in the scattering region from 105 - 155? I had to dig into the text to realize these are purely for visualization purposes and they don't represent actual measurements. The figures themselves could also be improved so it is easy to differentiate the differences between the different rows. E.g. by adding labels D_outer-envelope = 300 nm, and so on. Same comment for all other similar figures and their captions.

**Response:** In response to comment 4 of Reviewer 1 and in response to this comment, we have now indicated the point of difference within the panels of Figures 4-7, and we have now reworded the captions. In fact, as mentioned in response to comment 7 above, we have removed all of the aggregate labels from the manuscript and replaced them with descriptions of the aggregates. In addition, the following sentence has been added to the captions of Figures 4-7, "The range $\theta_{sca}$ = 135±20° of the calculations is highlighted on

each curve with a thicker curve." (We have also changed the color scheme of the curves to the CUD color scheme in https://jfly.uni-koeln.de/color/#convert.)

11. L285: The term 'side/back scatter' is rather imprecise and it shows its limitations in sentences like this. The precise quantity being considered is actually defined in Eq.7 and used in a number of the tables. I think the text would be improved by introducing and using a precise term for this quantity.

    **Response:** We have now replaced the term "side/back scattering" by "scattering at scattering angles 135±20°" in all places in the text where we are referring to the exact measurements and calculations that we conducted. We left "side/back scatter" in just a few places where we intend a more general meaning.

12. Sections 3.4 and 3.5: It is good the sensitivities to these different parameters have been explored. However, the readibily and conciseness of the text would be improved greatly by moving some of the discussion, figures and tables in these sections to a supplementary information file.

    **Response:** Reviewer 1 made a similar suggestion about moving some results to an appendix. As mentioned in our response to comment 9 above, we have now moved all of the tables of efficiencies and cross sections from all of the sections to Appendix B.

13. L656: In light of the many potential reasons for the discrepances between calculated and measured S/P ratios, many of which are nicely summarized in Section 4, I don't think it's reasonable to place particular emphasis on this one specific explanation here and in the abstract (as noted above).

    **Response:** Based on this comment, we have removed the sentence. "Given this, it is possible that models of scattering by BC aggregates that employ the random orientation assumption/option may underpredict the linear depolarization ratio of actual BC aggregates, and this should be investigated further in the future" from the Summary and Conclusions.

---

## Author Response (AR2)

**Editor's suggestions for minor technical revisions**

I would like to suggest some minor technical revisions, as follows
line 556: please simplify the sentence removing the two expressions of the ratio in the text and
leaving the results of the operations ($2.766 \times 10-2$ and $5.103 \times 10-1$)
Tables: verify number of decimal digits in tables
Table 6: "Range of values of" can be removed from column headers

**Response:** We have completed all of the above.